



# Integration of terrestrial observational networks: opportunity for advancing Earth system dynamics modelling

Roland Baatz[1], Pamela L. Sullivan[2], Li Li[3], Samantha Weintraub[4], Henry W. Loescher[4,5], Michael Mirtl[6], Peter M. Groffman[7], Diana H. Wall[8,9], Michael Young[10,19], Tim White[11,12], Hang Wen[3], Steffen Zacharias[13], Ingolf Kühn[14], Jianwu Tang[15], Jérôme Gaillardet[16], Isabelle Braud[17], Alejandro N. Flores[18], Praveen Kumar[19], Henry Lin[20], Teamrat Ghezzehei[21], Henry L. Gholz[4], Harry Vereecken[1,22], and Kris Van Looy[1,22]

[1] Agrosphere, Institute of Bio and Geosciences, Forschungszentrum Jülich, D-52425 Jülich, Germany
[2] Department of Geography and Atmospheric Science, University of Kansas
[3] Department of Civil and Environmental Engineering, The Pennsylvania State University
[4] Battelle, National Ecological Observatory Network (NEON), Boulder, CO USA 80301
[5] Institute of Alpine and Arctic Research, University of Colorado, Boulder, CO USA 80301
[6] Environment Agency Austria – EAA, Dept. Ecosystem Research, Spittelauer Lände 5, 1090 Vienna, Austria
[7] City University of New York Advanced Science Research Center at the Graduate Center, New York, NY 10031 USA
[8] Department of Biology and School of Global Environmental Sustainability, Fort Collins, CO USA 80523-1036
[9] Scientific Chair Global Soil Biodiversity Initiative, https://globalsoilbiodiversity.org
[10] Bureau of Economic Geology, University of Texas at Austin, Austin, TX, USA
[11] Earth and Environmental Systems Institute, The Pennsylvania State University
[12] CZO National Office, US NSF Critical Zone Observatory (CZO) program
[13] Helmholtz Centre for Environmental Research – UFZ, Dept. Monitoring and Exploration Technologies, Permoserstr. 15, 04318 Leipzig
[14] Helmholtz Centre for Environmental Research – UFZ, Dept. Community Ecology, Theodor-Lieser-Str. 4, 06120 Halle, Germany
[15] Ecosystems Center, Marine Biological Laboratory, MA 02543, USA
[16] Institut de Physique du Globe de Paris (IPGP), Sorbonne Paris Cité, University Paris Diderot, CNRS, Paris 75231, France
[17] Irstea, UR HHLY (Hydrology-Hydraulics), Lyon-Villeurbanne, France
[18] Department of Geosciences, Boise State University, Boise, ID 83725 USA
[19] Department of Civil and Environmental Engineering, and Department of Atmospheric Science, University of Illinois, Urbana, Illinois
[20] Department of Ecosystem Science and Management, The Pennsylvania State University
[21] Life and Environmental Sciences, University of California, Merced, USA
[22] Scientific Coordination Office International Soil Modelling Consortium ISMC, https://soil-modelling.org

*Correspondence to*: Roland Baatz r.baatz@fz-juelich.de



**Abstract.** Advancing our understanding of Earth System Dynamics (ESD) depends on the development of models and other analytical tools that integrate physical, biological and chemical data. This ambition of increased understanding and model development of ESD based on integrated site observations was at the origin of the creation of the networks of Long Term Ecological Research (LTER), Critical Zone Observatories (CZO), and others. We organized a survey to identify pressing gaps in data availability from these networks, in particular for the future development and evaluation of models that represent ESD processes. With the survey results, we look for gaps between data collection and ESD model development, to draw perspectives for the improvement both in data collection and model integration.

From this overview of model applications gathered in the context of LTER- and CZO research, we identified three challenges: 1) Improving integration of observational data in Earth system modelling, 2) developing coupled Earth system models, and 3) identifying complementarity and ways to integrate the existing networks. These challenges lead to perspectives and recommendations for strategic integration of observational networks and links to the ESD modelling community. We propose the further integration of the observational networks, either by 1) making the existing site-determined networks also functional topological networks with organising spread and coverage, and/or 2) thematically and geographically restructuring or co-locating the existing networks, and further formalizing these recommendations among these communities. Such integration will enable cross-site comparison and synthesis studies, which will offer significant insights and extraction of organizing principles, classifications, and general rules of coupling processes and environmental conditions.

## 1    Introduction

In light of accelerating global change (e.g. IPCC, 2014; Camill, 2010), the scientific and societal imperatives are paramount to improve our understanding of Earth System Dynamics (ESD), which comprise complex interactions among rock, soil, water, air, and living organisms that regulate the natural habitat and determine the availability of life-sustaining resources for human well-being (MEA, 2005). Understanding and modelling of Earth system processes and interactions among Earth system compartments can be enhanced by accessing a wider range of both observational and experimental data (Banwart et al., 2012;Aronova et al., 2010;Reid et al., 2010). For this purpose, observational networks have been developed the last decennia aiming at a temporal and multidisciplinary coverage of continental and global scale ecosystem observations. The Critical Zone Observatory network (Brantley et al. 2015), the International Long Term Ecosystem Research network (Mirtl et al., 2017), the National Ecological Observatory Network of the US (Loescher et al. 2017, Schimel et al. 2001), the Chinese Ecosystem Research Network (Fu et al., 2010) and the Australian Terrestrial Ecosystem Research Network (Lindenmayer, 2017) are a few exemplary networks that focus at the continental to global spatial scales, and daily to decadal temporal scales. One overarching goal of these research and observational networks is to use measurement data to improve the predictive capabilities of current models (Loescher et al., 2017). The growing availability of data (Hampton et al., 2013)



and an improved representation and resolution of modelled processes drive the development of Earth System Dynamics (ESD) models, with ever increasing sophistication (Wood et al., 2011), and novel validation and assimilation techniques (Penny and Hamill, 2017). Terrestrial Earth system models represent a large range of processes from tracking the fluxes and storage of energy, water, sediments, carbon, and other elements (scalars) as well as distributions and roles of organisms, land

use practices, climate, and humans (Mirtl et al., 2013). However, the majority of these models focus on one or a few processes, while integrated or coupled models offer frontier opportunities to explore cross-scale and cross-disciplinary processes that are needed to fully predict ESD responses to perturbations in combined local to global driving forces. No single ESD model can accomplish the full representation of pressure and response functions. An improved understanding of interactions and feedbacks among water, energy and weathering cycles with biota, ecosystem functions and services guides

the development of more integrated terrestrial Earth system models (Vereecken et al., 2016b). For this reason, coupled model suites and frameworks (e.g. Peckham et al., 2013;Duffy et al., 2014) are one option to expand our current modelling capability to incorporate cross-disciplinary processes for improved prediction of whole ESD system-level understanding, as well as for policy and management decisions. The incorporation of hydrologic, meteorological, biogeochemical and biodiversity measurements from within and across sites into such coupled model systems is a key component to providing

the multi-scale/multi-process understanding that is needed to advance predictions of ESD responses to land use and climatic changes, and the ever-increasing natural resource demand.

Critical Zone Observatories (CZOs) provide essential data sets and a coordinated community of researchers that integrate hydrologic, geochemical and geomorphic processes from soil grain to watershed scales (Brantley et al., 2017). CZOs offer the lens through which the interactions among the lithosphere, pedosphere, hydrosphere, biosphere and atmosphere can be

brought into focus (Brantley et al., 2007;Banwart et al., 2012). CZOs examine how scalar mass and energy fluxes interact with life and lithology over geological timescales that see the transformation of bedrock into soils, and how the same coupled processes enact feedbacks with changing climate and changing land use (Lin et al., 2015;Brantley et al., 2016;Sullivan et al., 2016).

The US Long Term Ecological Research (LTER) network was created with the aim to provide the data and information

needed for long-term, integrative, cross-site, nation-wide principle investigator (PI)-based research to advance our ecological literacy and act upon solving societal grand challenges (Callahan, 1984). This US project quickly gained attention, and sparked the foundation of other national and regional LTER networks (e.g. China, Europe, Australia). This lead to the foundation of the global International LTER network in 1993 (ILTER) (Kim, 2006;Mirtl, 2010;Vanderbilt and Gaiser, 2017), which currently comprises 44 formal national LTER networks. ILTER provides the scientific expertise, a global-scale

site network, and long-term datasets necessary to document and analyze environmental change. Larger scale case study areas (Long-term Socio-Ecological Research Platforms (LTSER)) are designed to support research on long-term human-environment interactions and transdisciplinary approaches. Recognizing that the value of long-term data extends beyond use at any individual site, the global ILTER network aims at making data collected by all (I)LTER sites broadly accessible to other investigators (e.g. Parr et al., 2002;Breda et al., 2006;Vihervaara et al., 2013), enhanced by the standardized



documentation of measurements and sites (Haase et al., 2016). Site metadata and data are increasingly accessible via the DEIMS-SDR web-service (Mirtl et al. 2017).

As the above review illustrates, each of these networks was created in an effort to address recognized science questions and knowledge gaps in specific disciplines that led their creation (e.g., ecology, geology). The collection of data within each

network – the variables measured, the methods by which they were measured, and associated campaign activities leveraging these networks – have been designed to address these specific knowledge gaps. At the same time, although these networks were not specifically designed to promote it, the nature of the organizing science questions, Earth system variables being characterized, and methods used for measuring those variables, lead to opportunities for across-network synthesis and co-production of knowledge across these networks and disciplines.

Here, the International Soil Modelling Consortium (ISMC, https://soil-modelling.org/) conducted a survey with participation from the ILTER and the global CZO network (Critical Zone Exploration Network) to identify knowledge and functional gaps in data availability, in our ability to integrate models and the linkage of data-model usage. ISMC envisions integration of models from different disciplines of hydrology, biogeochemistry and ecology, to increase the understanding and awareness of ESD processes, especially when these processes underpin other processes (e.g., global and regional climate)

(Vereecken et al., 2016b). To this end, the survey brings information on the level of integration of model approaches that use data from the LTER and CZO sites.

With the survey, we examined the complementarity and disparity among the models used in the two networks, identified gaps in current data collection and how those data are used in models, and propose a path forward to bridge these gaps for improving both novel data collection and model integration in ESD science. Details on the survey results and analyses can be

found in the Appendix, while survey questions and responses on model applications are found in the Supplement. We note that the survey was completed on a voluntary basis, which implies that not all modellers working with LTER/CZO data and models responded; thus, we acknowledge the list of models used in LTER and CZO communities is far from complete. However, more than 70 responses to the survey allowed exploration of the availability and use of observational data for modellers. Moreover, the survey informs the scientific community on the status of evolving and optimizing the monitoring

networks, and communicates individual and common values of these networks for the Earth system modelling community. The survey listed 52 variables (see Appendix Table 1) based on the common measurements in the LTER and CZO networks. The respondents were asked to identify the source of the data used, being data measured at sites (LTER and/or CZO) or obtained from other sources such as remotely sensed, modelled, or literature.

Based on the results of the survey, we come to define challenges and perspectives for 1) usage of observatory data in

integrated ESD models, 2) model integration in relation to specific disciplines, and 3) complementarity and possibilities for network integration.



## 2    Challenge 1: Data-model linkage

### 2.1    Current status

In response to the survey (see Appendix and Supplement for details), the variables used most frequently in models applied by CZO and LTER communities were from the atmospheric compartment (precipitation, air temperature, incoming

shortwave radiation, humidity, wind speed/-direction, and eddy flux of evapotranspiration and $CO_2$), followed by soil characterization (structure, texture, water content), above-ground biomass, and vegetation structure and dynamics. This reflects the current most frequent model requirements and applications in terrestrial Earth system science for coupled hydrological-biogeophysical models. The average model used 14 variables of the supplied list, ~2/3 of the variables for model input and ~1/3 for calibration/validation (Appendix Table 1). While the CZO model applications use variables and

data of saprolite and bedrock mineralogy, data on biotic and biodiversity variables were used more frequently in models associated with LTER. Models associated with CZOs applied significantly (Fisher test, $p<0.05$) more data of eddy flux measurements (evapotranspiration and $CO_2$), root density, soil water content, soil temperature, bedrock, and soil texture and physics compared to models associated with LTER. Models using data of habitat mapping, biotic and biodiversity elements were associated much stronger with the LTER community.

The majority of variables used in the models are sourced from on-site measurements (55% on average). The rest of the data (45%) is derived from other sources, mostly remote sensing (e.g., MODIS), to a lesser extent modelled (e.g., North American Land Data Assimilation System), external database (e.g., FluxNet), or literature sources. The most common remotely sensed variables were of the biosphere, especially habitat mapping, leaf area index, vegetation structure and dynamics, above-ground biomass, but also include snowpack distribution and duration.

Figure 1 presents the spatial and temporal scales modelled with the density of model applications. There is a large congruence in spatial and temporal scale between both modelling communities. Spatially, the model applications extend mostly from site- to catchment scales, with few occurrences at the lab or global scale. The high density in the center of Figure 1 shows the focus for sub-catchment scale modelling, and time scales of days to years, potentially decades, which is in line with the aims and conceptual basis of both LTER and CZOs. The specific inset diagrams show that for LTER, the

prevailing yearly, potentially decadal, time span is mostly covered at the site scale, whereas CZO associated responses work pre-dominantly at the catchment scale and daily time scale. Some CZO models seem to cover a larger range of spatial scales since the models indicated in the survey cover the full spatial range from macropore to continental and global scales. In terms of the modelled time scale, the long-time scales (centuries to millions of years) are mostly covered by models employed in the CZO community. This also lines well with the focus on management and predictive applications for LTER

associated responses, while CZO applications focus more strongly on the description of the interplay between various processes.



## 2.2    Example of data-model linkage from an LTER site

Next to the common, cross-site measurements, observation network datasets often also include some specific elements for individual sites, gathered from the perspective of prediction and model development, to deal with specific questions of ecosystem responses to global change. An example is the vegetation dynamics modelling in the French LTER Zone Atelier

Loire (Van Looy and Piffady, 2017), which uses predicted hydrological changes in river flow regimes and droughts to predict changes in land use and vegetation in the floodplain. It enables the construction of population dynamics models for characteristic tree species Black poplar and White elm for the LTER site where count data of the species populations is present. The proposed adaptation management scenario of water retention and restoration of flow regime and floodplain inundation proved successful according to the model to mitigate predicted climate change impacts on population dynamics.

Another example of vegetation dynamics modelling using observation network data concerns forest dieback under climate change (Breshears et al., 2005). At an intensively studied site of the Drought-Induced Regional Ecosystem Response Network, after 15 months of depleted soil water content, > 90% of the dominant, overstory tree species died. This combination of detailed spatial-temporal observational data on tree condition, soil water content, precipitation and

atmospheric conditions (temperature) allowed for data-driven development and validation of a regional model on drought-induced vegetation changes.

## 2.3    Open issues and implications

The LTER/CZO network sites monitor a wide range of environmental variables with long-term or at least regularly repeated

measurements, expected to provide more reliable and robust results than single measurements that produce 'snap-shot' information only. The application of long-term monitoring data in predictive modelling provides a strong opportunity in the era of ESD modelling (Parr et al., 2002). Application of the rich data collected at LTER/CZO network sites should improve process understanding and enable the scientific community to address the challenges of validating Earth system models that integrate coupled processes. Although holistic integrated models at LTER and CZO sites are used to raise understanding of

the coupled processes, cases are mostly restricted to individual sites and too limited in numbers. As the survey results have shown, for some themes (e.g., habitat/vegetation/crop) potential data from on-site field measurements are often replaced by remotely sensed or existing database information (Appendix Table 1). This suggests a strong need of on-site measurements for these data for modelling site-specific processes because on-site measurements generally are more accurate than remotely sensed or modelled data. Plausible causes for the lack of on-site measurements are the absence of harmonized measurements,

and the time and personnel-consuming requirements for data gathering.

In line with the observatory missions, models using LTER data more strongly embrace management and predictive applications (Stoll et al., 2015) ranging predominantly from site and sub-catchment to region, from days to decades,





compared to CZO models that focus more on process/system understanding and are more geophysical and subsurface oriented with a wider range of scales from lab to global, from seconds to millennia (Appendix Figure 6). The biotic theme in the survey is only covered by applications of LTER, whereas petrology and mineralogy are only covered by CZO model applications (Supplement). The results reveal the strong complementarity of models used in the CZO community addressing

below-ground processes while for LTER the focus lies more on above-ground processes.

## 3    Challenge 2: Model integration

### 3.1    Current status

To answer the question how data-rich LTER and CZO sites are used in holistic, integrated models, a 'level of integration' was calculated for each model by normalizing to the scale 0 (low) to 1 (high) the number of disciplines (e.g., atmospheric

circulation, biodiversity, biogeochemistry, and others, see Appendix Figure 6) and compartments (e.g., Atmosphere, Hydrosphere Lithosphere, and others) indicated by the respondents, and equally weighing them. The models using data from the LTER and CZO networks cover a wide range of integration, from very specific singular process models (little integration and variables), to highly integrated models that cover many disciplines and compartments, and have large numbers of variables. On average, a rather high level of integration is present for the models, meaning that the LTER and CZO data

model applications cover on average multiple disciplines (mean=3.6; sd=2.1) and ecosystem compartments (mean=2.7; sd=1.6). Models using data from both networks were in the higher data use range. Highest integration of data and disciplines in models is present in hydrological and geophysical disciplines. Model 'level of integration' is strikingly similar in LTER and CZO data applications and correlated strongly to the number of variables ($R^2 = 0.47$) (Figure 3).

### 3.2    Example of integrated modelling

Plant-soil interactions are changing across the globe, whether it be the encroachment of woody species into polar, alpine and temperate grassland areas (Archer et al., 1995;Jackson et al., 2002), the increase in atmospheric $CO_2$ concentrations that potentially alter the depth penetrations of roots (Bond and Midgley, 2012;Van Auken, 2000), or changing land cover (agriculture, forest plantations; (Van Minnen et al., 2009)). Subsurface changes to the root system architecture (root function,

density and depth) alters the injection (spatial distribution) of organic carbon into the ground, controlling microbial productivity and respiration, macropore location, distribution and evolution, controlling the transport of most water that moves through soil (Beven and Germann, 1982), and spatial distribution of organic acids and root respiration (generation of $CO_2$; (Jones, 1998)). These factors will impact infiltration of meteoric water charged with carbonic acid ($H_2CO_3$), influencing the breakdown of minerals and the redox conditions under which metals can be mobilized.

To explore the larger scale consequences of changes in root system architecture on soil water and riverine chemistry requires coupling of models from two scientific fields, a reactive transport model with a land-surface and hydrologic model to the



RT-FluxPIHM Model (Figure 4) (Bao et al., 2017;Li et al., 2017a). To further understand how these systems evolve in response to land cover change, we can couple these models with ecosystem models, such as Biome-BGC (Thornton et al, 2005). One limitation to such complex numerical models is the numerous datasets needed for parameterization. However, working with datasets derived from LTER, CZOs and NEON allows evaluation of model performance against data that

characterizes key processes embedded within integrated models. For example, RT-FluxPIHM is being used to examine the hydrologic and biogeochemical ramifications of woody encroachment into grasslands at the Konza Biological Station (KS, USA), a well characterized and monitored LTER site. Preliminary numerical experiments explore how differences in rooting depth and macro-porosity distribution (vertically and horizontally) alters groundwater flow patterns, and thus stream water discharge and solute behaviour (Figure 4b). These types of coupled models offer a way to explore plant-water-

biogeochemical feedbacks at the watershed scale and help guide future field experiments. As another example, the *Dhara* model resolves coupled soil-vegetation dynamics affecting water and biogeochemical processes. It has been used to examine the impact of tile drains on nutrient transport in agricultural watersheds (Woo and Kumar) and potential impact of ecohydrologic response to climate change on spatial hydrologic patterns (Le and Kumar).

**3.3     Open issues and implications**

Fragmentation and lack of integration has limited our abilities to understand the formation and function of ESD at various spatial scales, and to predict system response to global change and interaction of processes and parameters from local to continental scales (Grimm et al., 2013). Simple models, also called parsimonious models, are models which aim at predictions using as few variables as possible and limitting process representation to the necessary minimum. Parsimonious

models can provide good conceptual frameworks, also for integrated questions. At the same time, parsimonious models may not be spatially explicit but developed for highly specialized process representations and particular purposes. Therefore, parsimonious models stand in contrast to the desired prediction capability of models for nonlinear environmental system responses to global change. However, the development and use of parsimonious models is advantageous for targeted processes within disciplinary boundaries, yet, the applicability of conclusions drawn is highly limited (Basu et al., 2010;Li et

al., 2017b). It is an interesting question whether data access limits model integration, or whether a lack of integration is mainly caused by the model development side. Model integration is data-demanding. The numerical model applications necessary to test LTER and CZO conceptual model assumptions are integrated, process-based, spatially explicit models at the watershed scale that predict emergent behavior. The high level of multidisciplinary model inputs requires numerically expensive models and more importantly a sharp learning curve of the users. In our survey 10% of the models already used

data from both LTER and CZO networks in model applications. This implies that these models already integrated processes of interest to the two communities. It does not necessarily mean that the data are co-measured. Where CZOs mainly focus on understanding near surface structure and dynamics, integration with LTER might fill many of the ecological gaps in CZO studies by providing the scientific expertise, research platforms, and datasets necessary to analyze environmental change



with a particular focus on ecological driven processes. Whereas the conceptual models for LTER and CZO sites orient to a minimal level of integration, fundamental process understanding, with specific parsimonious models, nevertheless remains an essential part of the mission.

## 4     Challenge 3: Complementarity and/or disciplinary segregation

### 4.1     Current status

In an ordination of the model variables (see Appendix), models that used data of both LTER and CZO sites cluster in the center of the ordination of the surveyed models (Figure 5), which indicates that those models use a fairly similar and large group of variables. A significant number of models are co-located in the center, focusing mostly on hydrology and geophysical processes. The Group-A models associated with CZOs concentrate in the second quadrant, distinguished by the focus on modelling process in the petrology, regolith and bedrock. The Group-B models associated with LTER data are mostly located in the first and fourth quadrants (Figure 5) resorting under the ecology discipline related to habitat and biota. Disciplines separate from physics-oriented to biotic-oriented models, along with compartments from below-ground (negative) to above-ground (positive). The vertical axis distinguishes strongly integrated models, mainly hydrology-based, from specific rock weathering models focusing only petrology/mineralogy.

### 4.2     Open issues and implications to join the networks and communities' expertise

The survey revealed that models using data from both CZO and LTER networks generally cover a larger range of variables compared to models applied to only one network. Although few up to now, these examples show the opportunity these observation network data offer to integrate a larger number of compartments in models. Biotic data are applied in models of the CZO community (for instance, see Data Integration example), but they are few in number. Nevertheless, there is strong evidence for the important role of biota in coupled processes such as energy, carbon and nutrient cycling, and weathering (Filser et al., 2016;Richter and Billings, 2015;Wall et al., 2015). The outlier position of models focussing on biota and habitat variables in the ordination (Figure 5) indicates the need for integration of the biotic compartment in model development (Vereecken et al., 2016b). Therefore, harmonization and standardization of biotic observations is particularly important for those biotic variables related to processes and feedbacks with the hydrologic and biogeochemical cycles. In this survey, many biotic models were opposed to below-ground compartments in the ordination (Figure 5). This demonstrates the lack of data application for biotic processes in the subsurface, e.g., the representation of the weathering microbiome or root system architecture and dynamics (Smithwick et al., 2014). Recent initiatives address the missing integration of below-ground biota in the terrestrial Earth system science and models (Key to Soil Organic Matter Dynamics and Modelling – Keysom-Biolink project, Filser et al., 2016), along with the provision of substantial datasets on soil biota and biodiversity (global soil biodiversity database, Ramirez et al., 2015). Models *are* being developed that are capable of





estimating the role of biotic activity in soil formation, decomposition/mineralization processes, and predicting the carbon and nutrient cycles in specific soil types (Komarov et al., 2017;Wieder et al., 2017). Nevertheless, joining the often discipline specific data, as well as the largely site-to-catchment based but discipline specific modelling expertise of the CZO and LTER communities would lay the ground for new findings.

## 5    Outlook

### 5.1    Satisfying cross-disciplinary data demand for ESD models

The relationship of models and data is a relationship of mutual benefits. Data are collected in tremendous efforts for serving scientists to develop and test hypotheses (e.g. Braud et al., 2014;Clark et al., 2011). Models may also help scientists to better

design data collection strategies and tactics in observation networks (Brantley et al., 2016). With the increase in computational capabilities, stochastic methods such as data assimilation, global sensitivity and optimization algorithms are becoming more widely used. Commonly, these methods are used for parameter and state estimation. Additionally, stochastic analyses open the way to determine the observation requirements to reduce model uncertainties and answer hypotheses. Stochastic analyses can be used to identify key physical processes and their impacts if variables are subject to change.

Models can feed back to data through evaluation of improved observation strategies including process sensitivity to observation variables, parameters, measurement frequency, or resolution/extent in space and time (Lin, 2010). Considering observability, predictability, and the impact of heterogeneity on processes at the relevant scales through modelling could reduce resources spent on specific data gathering and benefit the gain in observing complementary data sets. Another aspect brought by merging data and modelling through data assimilation is the potential facilitation of upscale cascading of

knowledge from smaller-scale process understanding to larger-scale simplified representation, patterns and parameterization (Heffernan et al., 2014;Vereecken et al., 2016a). In this way, models can feed back to data and even drive observation requirements for maximum benefits for the model.

In particular, the reanalysis concept addresses the benefits of a strong model-data linkage. Frequently used in meteorological models (e.g., Compo et al., 2011;Dee et al., 2011), reanalysis is not used in terrestrial Earth system science, especially

ecology, yet, as it is in atmospheric science. For performing reanalysis, a physics-based model is fed with observations through a data assimilation scheme over a sufficiently long time period to update model states and parameters over time. Model states and parameters are optimized with the data assimilation method based on the observation, considering observation-, model- and forcing uncertainty. Application of reanalysis in Earth system models could generate gap filled, holistic, and coherent physics-based time series of terrestrial states, fluxes and parameters including variables characterizing

biological processes and biodiversity. Based on often non-continuous and sparse in-situ observations from long term observational networks such as CZOs and LTER, critical zone- or ecosystem reanalysis would need to specifically target biological and biodiversity related processes in Earth system models. The generated continuous reanalysis data could inform



further modelling processes or be used to test existing bio-geochemical concepts, hypothesis, and observations, and explore historical developments in the context of Earth system science. However, such ecosystem reanalysis depends on the quality of the observational data and needs to be physically coherent. Many challenges such as the optimal choice of the Earth system model, the data assimilation method, model parameterization and forcing data, validation data, and ultimately the

representation of biotic and abiotic processes need to be addressed for generating ecosystem reanalysis data encompassing variables characterizing biological processes and biodiversity.

Where there is strong evidence for the specific role of biota in ESD processes (Deruiter et al., 1994;Richter and Billings, 2015), there is nevertheless a general knowledge gap for biotic elements in integrated models. More specifically, a clearer understanding of the physical and chemical processes that shape the environment in which biota respond to climate forcing

and shape the physical environment themselves is required based on a solid observational basis to detect and predict thresholds, since small changes in temperature and precipitation may cause a non-linear and irreversible response in ecosystem structure and function (e.g., forest die-back Breshears et al., 2005). The emphasis of CZOs on geophysical processes reduces the focus on biotic and ecological dynamics that drive much of the dynamic responses of the Earth system. Phenomena such as community assembly, evolution, the emergence of pests and pathogens, and colonization by invasive

species has important effects on ESD, but are not well represented in most CZO studies. New initiatives have been launched recently to integrate the biotic component in Earth system science and models (Filser et al., 2016). Measurements of biota in the subsurface (e.g., bacteria, fungi, roots), especially at depth, is expecting strong developments in the coming decade as is the modelling (Grandy et al., 2016). As explained in the section on network integration, harmonization and standardization of biotic observations could better facilitate access to biotic observations related to processes and feedbacks with the

hydrologic and geochemical cycles. Network integration would feed the LTER need for a deeper geoscience emphasis.

## 5.2    Integration of models

Integrated models covering different disciplines and compartments are needed to objectively raise process understanding and develop predictive capabilities on the effects of climate and land use changes on ESD. In our survey, the few land surface-

atmosphere integrated process-oriented models like PIHM and Parflow-CLM were exceptional in the level of integration and application of observation network data. Along with a few other examples (e.g., Boone et al., 2009;Lafaysse et al., 2017), land surface models are rarely used to model processes with an integrated approach embracing biotic and abiotic variables using LTER/CZO data. A stronger communication between land surface modellers and LTER/CZO communities would enhance the integration of in-situ observations in models, but is challenged by the continental-to-global focus of land surface

models versus the site-to-region focus of LTER/CZO observatories.

Yet, integration does not necessarily mean that individual models need to increase in complexity. In the realm of higher integration for inter-disciplinary understanding of feedback mechanisms across disciplines and scales, parsimonious models can be integrated in a larger model platform (e.g. Peckham et al., 2013;Duffy et al., 2014) to investigate these feedback



mechanisms over climatic and geographic gradients. A model platform was also a clear demand issued by the majority (80%) of the surveyed modellers. The respondents were less unified as to what services should be provided on such a platform. Integration of parsimonious models into an integrated process-based model could be one service under such a model platform. Development of those models often requires the understanding of organizing principles, classifications, and

general rules of coupling processes and environmental conditions (Sawicz et al., 2011;Sivapalan, 2003;Sivapalan et al., 2003). Such insights can be gained through cross-site comparison and synthesis studies of observation data across different sites under gradients of climate, earth surface characteristics (e.g., soil type, lithology, topography, vegetation), and human impact (e.g., agriculture, pristine, urban) conditions, which observation networks are well positioned to carry out. In turn, the need for calibration/validation data of integrated models can only be achieved when a well-constructed network of biotic and

abiotic observations make inter-disciplinary data sets available to the model community.

## 5.3    Strategies for network integration

With respect to investigating specific aspects of ESD, the interactions of biotic and abiotic processes as well as below-ground and above-ground processes are key links, where geosphere focussed research by CZO and ecology focussed

research by LTER could benefit observation and model integration wise. This also counts for the highlighted lack of integration of underground biotic elements, although the underground biota performs a crucial ecosystem functioning role (Deruiter et al., 1994;Wall et al., 2015). Data harmonization could be achieved based upon the concept of blending the conceptual frameworks of Ecosystem Integrity (Müller et al., 2000) and the Essential Biodiversity Variables (Pereira et al., 2013) as suggested Haase et al. (2018) and the Critical Zone approach (Chorover et al., 2015 and Brantley et al., 2016).

Drawing from the requirements of ESD modelling, and the apparent complementarities and synergies already lead to network integration efforts in several regions around the globe, basically aiming at the joint use of resource intensive observatories by more than one research community. The integration of existing research and observation networks can be accomplished by either 1) making the existing networks also functional topological networks (Brantley et al., 2017), or 2) thematically and geographically restructuring the available networks. For the first option, both LTER and CZO need to

reconsider their existing topological structure in view of filling knowledge and observation gaps to their relevant large scale scientific questions crossing climatic, geographic and disciplinary gradients. For the second option, a new network can be proposed optimizing the thematic and geographic setting to answer the fundamental questions at the basis of the individual research sites´ and observatories' conceptual models, as well as answering the Earth surface evolution and formation questions in their full spatial and geographical context over biomes and continents. The design aim of network restructuring

would ideally be a controlled site selection process to cover climatic, geographic and disciplinary.

As indicated above, the situation of LTER and CZO research as well as the level of network integration and collaboration vary strongly between countries and continents around the globe. The challenges ahead for US-LTER, US CZO and NEON are as specific as those for the European efforts to establish one joint research infrastructure. Considerations about network





integration also need to consider differences in the organizational structure, where CZOs have been mainly based on scientific networks and projects, while LTER has established formal governance structures in regional groups and globally. The levels of implementation and formalization range from regions with well-established networks (US-LTER, US CZO and NEON), to countries and regions where research and observation networks are based on the initiative of individual sites,

observatories or projects.

Existing Earth observation networks such as NEON already are systematic in topographical constitution, offering opportunities for the proposed geographical integration. A notable European initiative is the Integrated European Ecosystem, Critical Zone and Socio-Ecological Research Infrastructure (eLTER RI), which comprises the definition of overall research challenges, includes the focal aspect of CZO research and requirements of widely ecosystem models. In order to successfully

restructure the existing observatory networks, a dialog is necessary with the demand side of the ESD modelling community and to the need for integration of data with respect to the calibration/validation of current integrated models (e.g., Coupled Model Intercomparison Project, Meehl et al., 2005). This will help troubleshoot some essential missing parameters or problems with spatio-temporal resolution and measurement accuracy. Furthermore, this interaction is necessary to construct the topology of an integrated research and observatory network, based on basic questions in ESD process understanding over

larger scales and model calibration/validation needs to be answered by observatory network data.

## 6   Concluding recommendations

The CZO and LTER networks should act as synthesizers of interdisciplinary research approaches that lead to emergent understanding and ultimately result in more deeply-informed process-based models (Brantley et al., 2017;Rasmussen et al.,

2011), in particular crossing the boundary of geoscience that guides CZO and bioscience that guides LTER. To be effective for this objective, a stronger dialog is needed between the observatory networks. With the modelling communities, better communication is needed to enhance integration of data in modelling and thus strengthen the crucial role of the observatory networks in raising understanding of ESD processes and derive prediction capabilities for drivers, impacts and responses to global change. Additionally, there is a need for more conversation/coordination between modellers and empiricists to

develop strategies for observation networks. The role of discussion amongst stakeholders, decision makers, funding agencies, observatory networks, and the broader scientific community cannot be over-stressed. The rapidly increasing technological capabilities in computational power, ground based instrumentation, and unmanned automated remote sensing technologies require all stakeholders to decide on which aspects the future observational requirements shall focus. Given today's grand challenges, the communities need to focus on expanding observation efforts/targets within allocated resources

thanks to cross-community harmonized methods and data sets. The communication and exchange about services, tools for data-model linking through web platforms offers obvious opportunities in this sense.



Finally, we can state an essential need to educate and train the next generation of Earth system scientists for modelling capabilities across disciplines. This indicates the need for dedicated Earth system science university courses, online teaching materials on model use, and a coordinated, community driven modelling platform.

**Acknowledgements**

Financial support from NSF, NEON, LTER and CZO made this collaboration possible. HWL and SW acknowledge the National Science Foundation (NSF) for ongoing support. NEON is a project sponsored by the NSF and managed under cooperative support agreement (EF-1029808) to Battelle. Any opinions, findings, and conclusions or recommendations expressed in this material are those of the authors and do not necessarily reflect the views of our sponsoring agencies. We
dedicate this work to the memory of Henry L. Gholz who was strongly involved in developing these strategies when the Rocky Mountains took him from us (RIP).

## 7    Appendix

### 7.1    Methods: Survey method and results

Questionnaires are a useful approach to derive opinion in a structured way from communities through individual responses
(e.g. Blume et al., 2017;Reiners et al., 2013;Steel et al., 2004). In order to investigate the use of models and linkage of models to data in the CZO and LTER communities, a questionnaire was set up with multiple response questions. The communities were addressed by e-mail to principal investigators (PIs) of the CZOs through the Critical Zone Exploration Network (approximately 1200 individuals), and the PIs of the ILTER network (approximately 400 individuals) with the request to forward the survey to respective modellers. The first part of the questionnaire characterizes the computational
model used in respects such as associated network, geographic region, purposes of the modelling activity, spatial and time scale, compartments, disciplines and model structure. The second set of question inquired the type of variables/data used, purpose of the data usage (model input or calibration), and the source of the data used in the model (measurements at sites, remotely sensed, database, modelled or literature). The third part of the questionnaire investigated whether and for which specific purpose the respondents would desire a model integration platform. Except Yes/No questions, all questions were
answerable with multiple responses by simple ticks. The questions were designed through iterative feedback loops together with leading scientists of earth system disciplines.

### 7.2    Methods: LTER and CZO variables

The specificities, purpose and scientific origin of the LTER and CZO networks accumulated to the design of a suite of
variables and parameters desirable and common to be measured at the respective sites. In total 52 variables were surveyed in this survey, based upon the ecosystem integrity concept (Müller et al., 2000), the essential biodiversity variables (Pereira et



al., 2013;Haase et al., 2018), the common CZO measurements (Chorover et al., 2015 and Brantley et al., 2016), and variables for energy and matter balances at the catchment scale. The ecosystem integrity concept formulated conceptual requirements to evaluate the state and integrity of an ecosystem (Andreasen et al., 2001). The intention of ecosystem integrity assessments was to provide information for knowledge based policy and management decisions for improved

ecosystem integrities. Haase et al. (2018) linked the ecosystem integrity concept with the essential biodiversity variables framework (Pereira et al., 2013) to formulate the most recent set of biotic and abiotic variables which are desirable to be measured at LTER sites around the globe. The list of common CZO measurements (Chorover et al., 2015) was formulated to allow for a site-wise comparison of infrastructure and measurements at the ten CZOs in the United States (Chorover et al., 2015). Each observatory was funded based upon very site specific questions, hypothesis and therefore set of measurements.

With the common set of CZO measurements and protocols, a foundation is laid to develop concepts how a network of CZOs can be established to address scientific questions across scales. Variables regarding mass and energy fluxes were in common by all three concepts, some biogeochemical parameters were specific to the common CZO measurements and some variables of the biota were specific to the biodiversity indicators list. The set of variables in this survey entails variables from all three concepts and was gathered with regard to the potential use, opportunities, complementarities and synergies of the variables

and both networks.

### 7.3    Methods: Statistical analysis

For analysis of the survey data, three statistical methods were used. Fisher's exact test (Agresti, 2002) was applied for testing whether variables or model characteristics were independent of the network associated with the respondent. The null

hypothesis of the Fisher's exact test was that the variable occurs independent whether the response is associated with CZO or LTER. Hence, it was tested whether particular variables or model characteristics can be associated for a specific network (rejection of null hypothesis) or if they are equally distributed over models of both networks to a signficance level indicated (0.95% or 0.99%). The Mann-Whitney U-test (Bauer, 1972) was applied for testing whether the distribution of model complexity (range: 0-1) and number of variables (range: 0-52) across the responses was distributed independently of the

associated networks (CZO, LTER or both). For analysis of trends in the variable space, the detrended correspondence analysis (DCA) (Hill and Gauch, 1980) was performed using the DECORANA package implemented in R and Fortran (Oksanen and Minchin, 1997). DCA ordinates presence-absence data, in this case, the response whether a variable was used in the model application (1) or not (0). First, a correspondence analysis is done by iteratively calculating the scores for the model responses as weighted average of the total variable abundance. Second, the scores are detrended and rescaled by

cutting the axis scores of the correspondence analysis into segments and moving the residuals into a secondary axis. As usual, only the first two axes will be used as those yield the strongest correspondences.



## 7.4 Results on model characteristics

118 surveys were returned with information on model application. 70 out of 118 surveys provided full information related to model characterization and variables. Nearly half of the respondents (47 %) reported CZO observational data application, two-thirds (66 %) associated LTER data to their model and 12 % model applications used data of both networks (Appendix Figure 6a). Geographically, the majority of model applications came from Europe (63 %), followed by North America (27 %), global (18 %), Asia (12 %), and Africa (5 %). Particularly in Europe, a large fraction of respondents were associated with LTER, while in North America the CZO community was the most responsive. The purposes of models were representatively distributed; however, LTER model applications were more strongly oriented to management/prediction, compared to CZO model applications that were relatively more focused on process and system understanding. In terms of the modelled time scales, the long time scales over centuries to millions of years are mostly covered by models employed in the CZO community (Figure 6d). Spatially, the model applications extend mostly from site- to catchment scale, with few occurrences at the lab (n = 3) or global (n = 5) scale. Compartment wise, CZO model responses worked more the lithosphere and cryosphere (Figure 6e). Except for biodiversity, the modelled disciplines are rather evenly distributed. Biodiversity is exclusively covered in models of the LTER community. Although models employed for CZO may include biota, this was not the case for the 22 models associated with CZOs in this survey. Since only few models were used by more than one respondent, the survey responses indicate the application of a large variety of models in terms of disciplines and scales (Figure 6).

The measured variables used most frequently in the models were from the atmospheric compartment (295 times in total), respectively: precipitation, air temperature, incoming shortwave radiation, humidity, wind speed/-direction, and eddy flux of ET and $CO_2$. The next most frequently used variables were soil structure (soil depth, layers), above-groundbiomass, instantaneous discharge, texture and physical characterization, soil water content, and vegetation structure and dynamics. In contrast to these, individual variables associated with biodiversity (60 times in total), and saprolite and bedrock properties (50 times in total) were much less used in the models. Most likely, this is because they are much more sampling intensive in terms of time and resources compared to atmospheric variables (e.g. a weather station). The variables used reflect also the current most frequent model applications in terrestrial Earth system science. CZO and LTER data are mostly applied for coupled hydrological-geophysical models being the most data intensive types of models, which may as well be reflected in these statistics. Biogeochemistry, petrology and biology communities appear as less data intensive.

As could be expected, the CZO models most frequently apply variables of saprolite and bedrock (54%, plus 26% joint CZO and LTER model applications) while variables associated with biodiversity appear more frequently in LTER model applications (91.7%, plus 8.3% joint models). The following variables are significantly (Fisher test, $p<0.05$) more frequently applied in CZO site modelling: 'eddy flux of ET and $CO_2$', 'root density', 'soil water content', 'soil temperature', 'bedrock





soil texture and physics'. Models using data of 'habitat mapping', and 'vascular plant diversity' were associated with the LTER community.

The majority of variables used in the models are sourced from on-site measurements (55% on average). 45% of the data is sourced from other sources, including remotely sensed, modelled, external database, or literature. The fraction of on-site measured data use is slightly higher for the variables in the atmospheric compartment (58%) compared to, for example, the fraction of variables represented by habitat/vegetation/crop. Compartment wise, the fractional use of empirical, site-based measurements is highest for the variables belonging to the surface waters (68 %). The most commonly remotely sensed data in the survey were variables of the biosphere, especially 'habitat mapping', 'leaf area index', 'vegetation structure (height) and dynamics', 'above-groundbiomass', 'snowpack distribution and duration', 'eddy flux of evapotranspiration', and 'CO2'. Model wise, the average model used 14 variables, 9 of those as model input and 5 of those for calibration/validation (Appendix Table 1).

The model 'level of integration' was calculated by normalizing the model-wise number of disciplines and compartments indicated by the responses to a scale of 0 to 1 (high) and equally weighing both to obtain the 'level of integration' between 0 and 1. On average, a rather high level of integration is present for the models; meaning that the CZO and LTER data model applications cover on average multiple disciplines (mean=3.6±2.1 standard deviation) and ecosystem compartments (mean=2.7±1.6). Model 'level of integration' is strikingly similar in CZO and LTER data applications. The richness in variables was positively correlated to the number of disciplines ($R^2 = 0.29$), and to compartments ($R^2 = 0.4$). However, unifying compartments and disciplines to the 'level of integration' measure correlated most strongly to the number of variables ($R^2 = 0.47$) (Figure 3). The level of integration for models used at CZO and LTER sites raises with the number of variables used. Models using data from both networks were in the higher data use range. In terms of purpose, models used to gain 'process understanding' were described by a significantly higher median of both *level of integration* and *number of variables*. The time scale of 'days' was also indexed by a high number of variables and level of integration.

The question whether the CZO and LTER modelling communities would benefit from a model platform providing data-model linking services, was answered affirmative by a large majority (80%) of the surveyed modellers. The respondents were less unified as to what services should be provided on such a platform. Most saw it primarily as a 'marketplace' with links to data platforms, some advocate that the model platform could even distribute software tools and models. With the promising observation that 10% of the models already integrated data from both networks, the need for more integration between the CZO and LTER communities is also underpinned by this expressed desire for an integrated model platform.



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



**Figures and Tables**

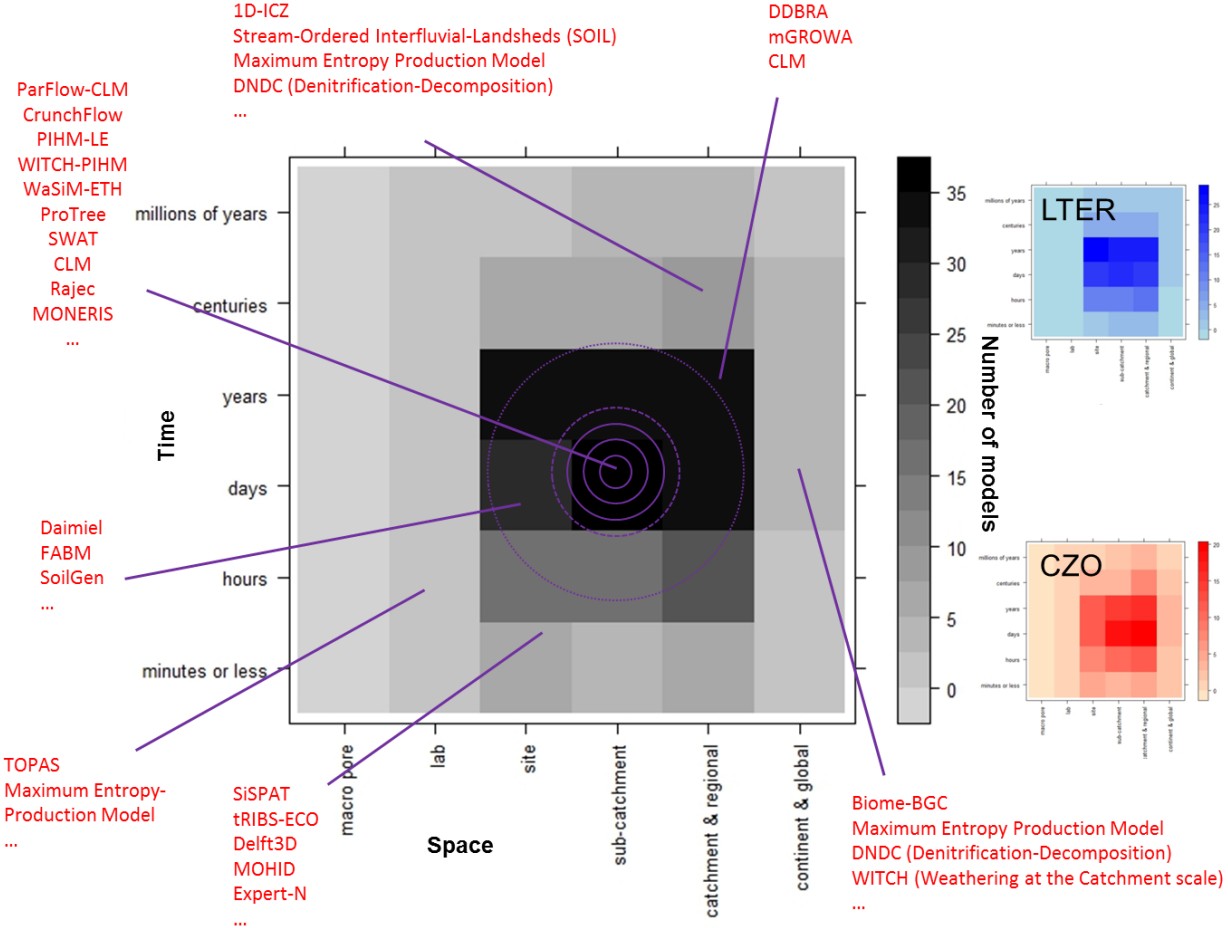

**Figure 1:** Heat map of model applications, with spatial and temporal dimensions of the surveyed models. Most models were denoted using several temporal and spatial scales. For visualization, we present some individual models at one exemplary instance. On the right, CZO and LTER model applications are presented separately.



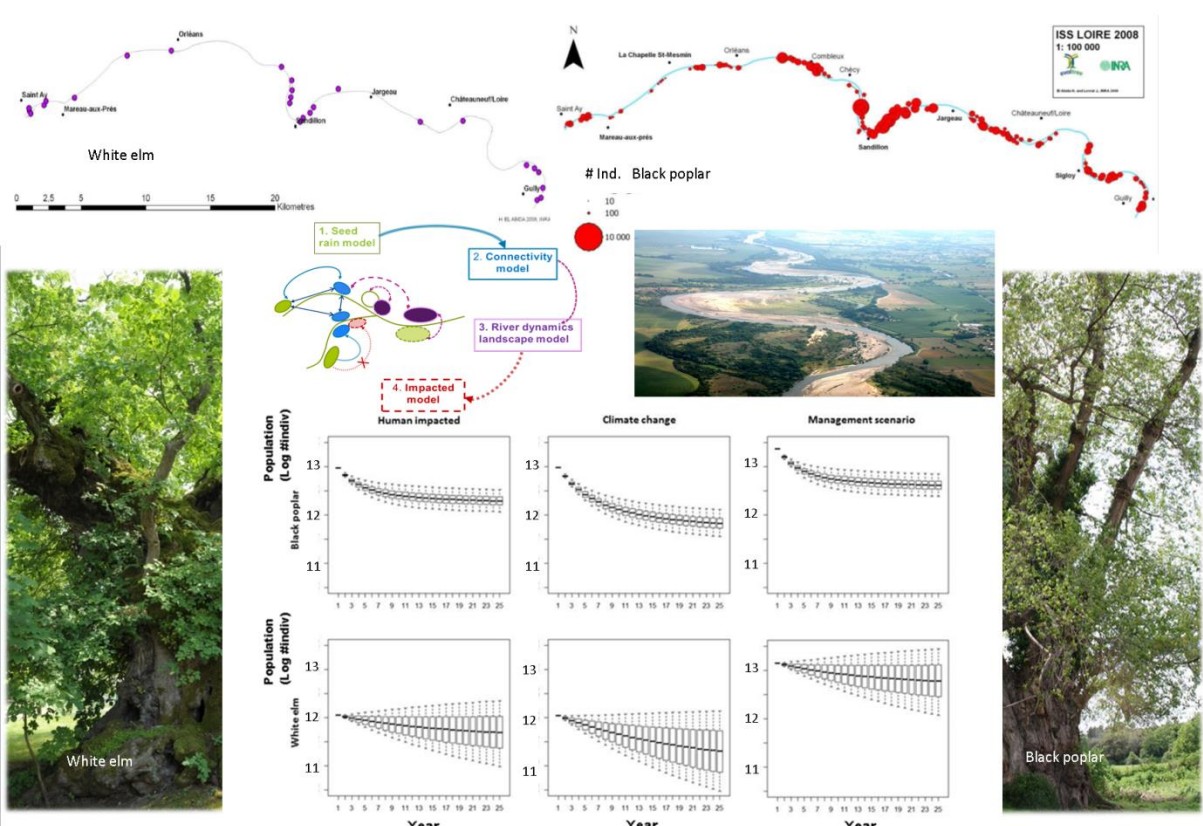

**Figure 2: Integrated model of vegetation dynamics (with population dynamics and landscape dynamics modules) in the French Loire LTER, constructed with data of tree individuals, flow regime, sedimentation and climatic conditions. Outcome presenting the total Loire basin population (logarithms number individuals) for 25 year in a baseline scenario (human impacted), projected climate change, and adaptation management for Black poplar and White elm (Van Looy and Piffady, 2017).**





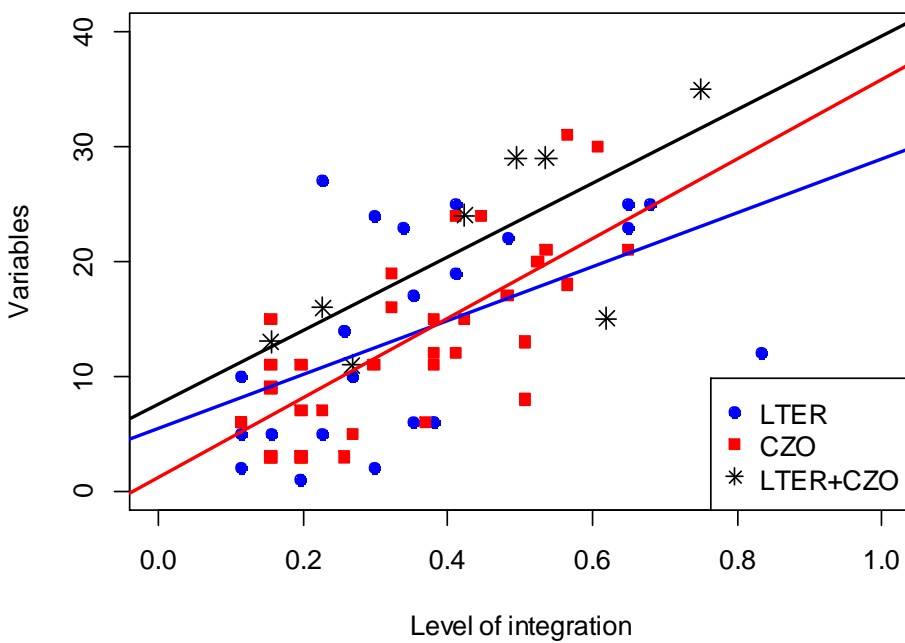

**Figure 3: Model-wise 'level of integration' calculated from the summed scientific disciplines and modelled compartments for the corresponding model. Trend lines corresponding to the models associated with LTER (blue), and CZO (red), and models associated with data from both networks (black) are presented.**





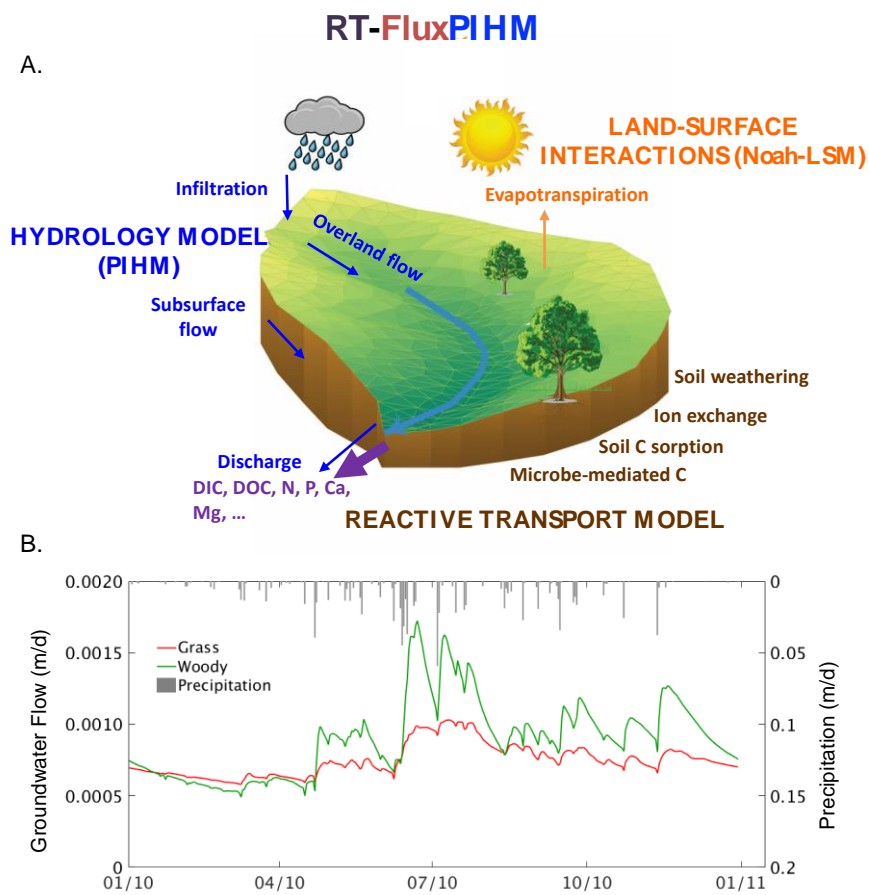

**Figure 4: A. Representative processes from different disciplines (different colours) in different modules that can be integrated into RT-Flux-PIHM (Bao et al., 2017). The integration among processes from mechanistic bases in the model will allow systematic**
5 **understanding at the watershed scale B. Simulated difference in groundwater flow for a grassland (red) and woody encroached (green) watershed (N04D), at the Konza Prairie, KS (USA). Grassland simulations are parameterized with a 0.3 m rooting depth and enhanced horizontal macropore development, while woody encroached simulations have a rooting depth that extends to 1.0 m deep with the enhanced vertical macropore development, all other parameters are the same between the simulations.**



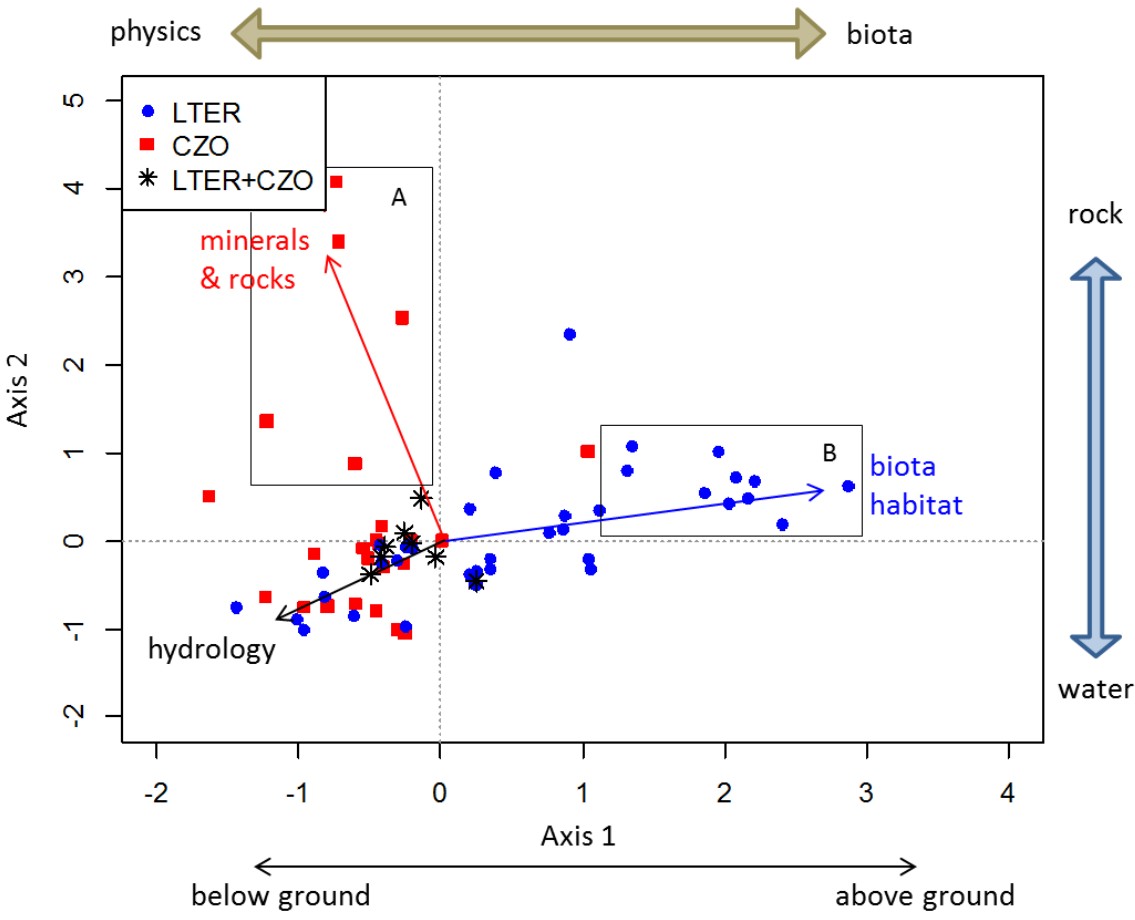

**Figure 5: Ordination of the models and data in the survey; clusters show significant differences for models of LTER and CZO. CZO cluster A focuses on variables of the saprolite and bedrock, LTER cluster B focusses on the variables of biota/biodiversity.**





**Figure 6: Distribution of respondents associated with either LTER, CZO, or both sites (a) to geographic region modelled (b), purpose of the modelling (c), time scale modelled (d), disciplines (e) and compartments (f) integrated.**





**Table 1: Summary table of variables, associated ecosystem compartment, the times it was associated with a model, the source for the data being either from site observation or from another source: i.e. remotely sensed, modelled, external database, literature.**

| Ecosystem compartment | Subcategory / theme | Variable | Sum of all instances nr. | Source:sites [%] | Source:others [%] |
|---|---|---|---|---|---|
| Atmosphere | | Eddy flux of ET, $CO_2$ | 35 | 52 | 48 |
| | | Air temperature | 47 | 62 | 38 |
| | | Humidity | 38 | 65 | 35 |
| | | Incoming shortwave radiation | 42 | 62 | 38 |
| | | Wind speed / -direction | 37 | 66 | 34 |
| | | Precipitation | 49 | 61 | 39 |
| | | Throughfall | 25 | 41 | 59 |
| | | Snowpack distribution and duration | 22 | 56 | 44 |
| Vadose zone | Solid phase | Elemental composition and mineralogy | 12 | 60 | 40 |
| | | Texture and physical characterization | 33 | 59 | 41 |
| | | Structure (soil depth, layers) | 35 | 61 | 39 |
| | | Organic Carbon | 24 | 61 | 39 |
| | | Radiogenic isotope composition | 2 | 50 | 50 |
| | Litter | Litter composition and biomass | 19 | 48 | 52 |
| | | Soil respiration | 15 | 50 | 50 |
| | | Microbial biomass above-belowground | 10 | 36 | 64 |
| | | Root density | 21 | 31 | 69 |
| | Liquid phase | Soil moisture | 32 | 57 | 43 |
| | | Soil temperature | 24 | 62 | 38 |
| | | Hydraulic head | 20 | 44 | 56 |
| | | Matric potential, specific conductivity | 24 | 45 | 55 |
| | | Water chemistry | 19 | 54 | 46 |
| Saprolite and bedrock | Solid phase | Texture and physics/structure | 18 | 45 | 55 |
| | | Element composition/organic matter | 8 | 67 | 33 |
| | | Petrology/mineralogy | 7 | 43 | 57 |
| | | Age or rate constraints (radionuclides) | 3 | 25 | 75 |
| | Liquid phase | Potentiometric head, temperature | 7 | 38 | 63 |
| | | Groundwater chemistry | 5 | 38 | 63 |
| | | Gas chemistry | 2 | 100 | 0 |
| Surface water | Hydraulics | Instantaneous discharge | 34 | 54 | 46 |
| | | Sediments | 17 | 62 | 38 |



| | | | | | |
|---|---|---|---|---|---|
| | Water quality | Water temperature, Electrical Conductivity, pH | 25 | 69 | 31 |
| | | Water quality – Spectral Absorption Coefficient (DOC) | 20 | 68 | 32 |
| | | Water Quality (nutrients, major cations / anions, others) | 29 | 63 | 37 |
| | | Stable isotopes | 9 | 90 | 10 |
| Biosphere | Habitat/ vegetation/ crop | Habitat mapping | 27 | 42 | 58 |
| | | Structure (height) and dynamics | 32 | 43 | 57 |
| | | Above-groundbiomass | 35 | 52 | 48 |
| | | Leaf area index | 27 | 46 | 54 |
| | | Photosynthesis (Chlor a) | 16 | 45 | 55 |
| | Biota, diversity | Birds | 4 | 80 | 20 |
| | | Ground beetles/spiders | 5 | 57 | 43 |
| | | Soil invertebrates/gastropods | 7 | 56 | 44 |
| | | Soil microbial diversity | 5 | 60 | 40 |
| | | Benthic invertebrates/fish | 6 | 57 | 43 |
| | | eDNA (environmental DNA; species detection) | 2 | 50 | 50 |
| | | Food web diversity (e.g. AMMOD) | 7 | 67 | 33 |
| | | Vascular plant diversity | 11 | 53 | 47 |
| | | Lower plant diversity | 7 | 50 | 50 |
| | | Fungi | 4 | 50 | 50 |
| | | Biofilm | 1 | 100 | 0 |