# Peer review of "Steering synergies in terrestrial observation networks: opportunity for advancing Earth system dynamics modelling"

_Earth System Dynamics, 2017_

## Short Comment (SC1) · 16 Nov 2017

I got interested in LTER and CZO sites locations. Usually, data series availability from tropical and subtropical climates are very scarce when compared with temperate climates. Thus, the balance between sites location in different climates may be a key factor for models integration and tell something about the expected simulation bias in ESM derived from these networks.

---

## Referee Comment (RC1) · Anonymous Referee #1 · 19 Dec 2017

The study by Baatz et al. assess the current status of the level of integration of models used by the LTER and CZO communities, and gives perspectives how the fusion of measurement and modelling communities could exploit the full strengths of observational networks to increase our understanding of Earth System Dynamics. In my opinion, this attempt is timely and deserves the presentation in a journal like Earth System Dynamics. There are many straightforward thoughts in this manuscript, and also the analysis of level of integration and the survey itself are useful instruments. However, I see three major points that need thorough revision.

1. The utilization of terms (integration, incorporation, linkage of data model-usage, coupling) is not precise, and especially terms like model integration should be defined

thoroughly and then remain reserved for the given used. One example is the abstract: "Advancing our understanding of Earth System Dynamics (ESD) depends on the development of models and other analytical tools that integrate physical, biological and chemical data. This ambition of increased understanding and model development of ESD based on integrated site observations was at the origin of the creation of the networks of Long Term Ecological Research (LTER), Critical Zone Observatories (CZO), and others". If I understand it correctly, in the second sentence, "integrated site observations" means a set of measured variables that comprises both driving variables and target (mapped) variables that are necessary to run and validate a process based model. As the preceding sentence already uses integrated, I suggest rephrasing the next one, otherwise the readers will stumble already here. In addition, objective 3) (Line 10) suggests network integration. This needs to be specified, and distinguished from model integration. In addition, the difference between model coupling and integration is not clear, and the terms are used interchangeably (see section3.2), though they are not: Is integration of a new variable more pertinent or could these processes be used at the same time, or would model coupling be more efficient? Furthermore, it remains unclear what exactly the authors mean with data-model linkage, and how was it quantified?

2. To this end, I suggest introducing a figure on how data assimilation, data integration and model coupling are related, where can they complement each other, and what are the differences?

3. The structure of the paper needs to be changed. A materials and methods section is required to give the reader an overview on the questions posed in the survey, and on the evaluation methods. The current Appendix structure (Appendix figures were missing) produces duplicate information which could be avoided when materials and methods section would follow the introduction.

In general, the different paragraphs even in a section or sub section need to be linked in a more straightforward way, and the writing styles need to be harmonized.

See some more detailed comments below.

Title

ok

Highlights

-

Abstract

L6: "we look for" sounds colloquial, I suggest changing to "The survey results identified gaps . . ."

L10: please specify complementarity. Unclear if you mean complementarity of networks or observations that occur at given sites, but are missing at the other.

L13: functional topological network: Formulation unclear. Please specify.

Introduction

L20: this is an odd sentence, I don't think it's verbatim. What exactly do you refer to with "scientific and societal imperatives" which are paramount to improve our understanding? Please rephrase the sentence, and probably split it.

L30: exemplary is ambiguous here. I suggest to change to "are for instance networks . . ."

P3L2: ESD has been introduced before.

P3L3: please rephrase sentence, it starts with ". . .range form . . .", but there is not "to".

P3L2-10: This section is fundamental in this contribution. It contains data assimilation, an attempt to define integrated/coupled models and calls for "more" integrated models. The authors need to clearly define the borders what can be achieved with data assimilation, model integration and model coupling, work out what the differences are and to comment on if "more integrated" models are less or more efficient than coupled models. I also suggest having a figure here on how data assimilation, model integration and model coupling is connected or distinguished.

[Figure]

L11: how exactly are more integrated and coupled models different? For more integration, capabilities of a usually couples model need to be implemented in the parent (less) integrated model. There are also several ways of coupling, which should here described in detail.

L13: At this point the authors bring in another terminology: incorporation of measurements. I suppose, this is in no way different from data integration. Please be very clear at this point. Incorporation also contradicts the coupling approach promoted in the preceding sentence.

L17, and following two paragraphs come out of the blue. A connecting, introducing sentence is required at this point.

P4L6-9: The sentence seems too long for the part of the message I understand. I am not sure what your mean with organizing science questions. If I get it right, I suggest: "At the same time, the science questions, observed variables and associated measurement methods lead to . . ."

L10: I suggest deleting the "Here".

L12: still at this point model integration and linkage of data-model usage remain vague.

L17-28: Introduction could stop here. I find it very odd that you dive in section 2 directly into the Challenges without describing what was asked in the survey. I strongly advocate for a materials and methods section, as the neglect of the named section leads to duplicate information in the appendix.

Materials and Methods (in appendix, unfortunately)

Results

P5L1: How is data-model linkage defined? This needs to be explained in a preceding section (e.g., Materials methods or such like)

P5L3: Appendix and supplement: quite vague. All appendix, all supplements? The fundamental data should occur in the main text body.

L15: Does "used" mean that it is an input variable or a target variable. Please

consider having the categories input and target variables at this point. It may be useful information if calibration/validation data has been determined at the site directly.

L20: I don't see the relation of this paragraph to the section head "data-model linkage". Please consider having another section head, or link figure one with (for instance) scales of the input data. This would be the link to your section head. In Figure 1 I suggest leaving the 0-values white.

P6, section 2.2: I suggest reducing this section to one paragraph, i.e., only giving one example, but explain this properly. The corresponding figure is illegible, especially the river-like structure, the model structure? (seed rain model and some ellipses and arrows, without explanation), is it boxplots I am looking at, is the reference scenario "human impacted" only? Also after several readings it is not clear to me what the difference between data-model linkage and integrated modelling should be, especially if the Figure caption starts with "Integrated model . . ." and section 3 is on integrated modelling. The reference to the figure is missing.

L24 "holistic integrated models". This class has not been defined before. Does this class comprise all the models used by the survey respondents?

L25/26: unclear what "replaced" means. Would modellers prefer using other products than the directly measured ones, and what would be the reason? The following sentence claims that there is a need for on-site measurements, but would they be used?

P7L9-10: All disciplines and compartments should be subject to the main text.

Subsection 3.1: I don't fully understand how the level of integration was calculated exactly, and how the variables (L12, 14) find their way into this metric. Please specify. The metric "level of integration" is also not discussed in this section, but there is only a reference to the figure.

L30-31: According to the subsection head this section is about integrated modelling. In the named lines, it says "requires coupling of models. . .". This switch of model integration to coupling is really confusing, and inadequate.

P8L1: RT-FluxPIHM: what is the purpose of this model?

L10: I suggest stopping here (don't give the next example) and tell the reader what the preliminary results were. The figure suggests increased groundwater flow though now trees with a greater rooting depth are present. Please discuss this briefly, and what regional implications does it have?

P9L18: please add the appropriate reference to your analysis.

P10L18-20: Please rephrase sentence.

P11L18-20: True, but this paper would be a chance to describe how this could be set into action.

P12L29-30: Please finish this sentence. This section: what do the communities think of your two suggestions? Would a new network replace the old ones? Please also specify the term functional topological networks. What does it imply for the networks.

P13L19: here the term "more deeply-informed" comes into play as another model class. It is necessary for the authors to clarify how this translates to model coupling, integration in their opinion.

---

## Referee Comment (RC2) · Anonymous Referee #2 · 16 Jan 2018

The manuscript discusses a very pertinent problem in Earth System Dynamics and for that reason merits consideration in principle. However, it does so without sufficient technical coherence and depth, transmitting the notion of a vague opinion paper than a thoroughly developed scientific study, which would be needed to provide solid grounds to the argumentation conducted in the manuscript.

The obstacles being already at a scientific language level, where there is no syntax unicity behind fundamental concepts. In practice, each and every term prone to cause confusion should be clearly defined in a rigorous manner, e.g. as the other reviewer pointed out "integration, incorporation, linkage of data model-usage, coupling". Oth-

erwise this is not a scientific paper, but rather a soft-formulated style of manuscript tailored for a non-scientific audience (which has its merits but is outside the scope of ESD).

While articulate divagations with catchy phrases and buzzwords may be popular in some soft venues in environmental science (e.g. Water Resources Research, WRR) and outreach communication (EOS or general media), this sort of approach feels out of place at ESD. In fact, here a rigorous "hard-science" treatment and discussion of problems should rather be the norm. For every bold statement of opinion, a rigorous scientific argumentation is a fundamental requirement, and in many cases mathematics are actually extremely helpful for that purpose.

My recommendation is thus to restructure and strengthen the paper so that its true scientific DNA comes out and the formulation, methodologies, definitions and overall concepts are all clearly stated. To drop the fashionable WRR-style beauty without substance, and rather take a more sober, theoretical ecology or ESD style scientific construct. The ESD readers will highly appreciate your insightful rigour.

Moreover, a clearer distinction should be made between the known facts and insights, and the novel contributions brought on by this study. The strategies followed in the study should also be more explicitly and rigorous formulated and justified, alongside with their merits and caveats.

At some stages, the reader is left to wonder what has actually been done in practice, and what are essentially suggestions and statements of purpose. The formal procedures belong to the main body of the manuscript, as they are fundamental to bring more substance to the eloquent argumentation.

In order to actually grasp some basic sense of the scientific work beneath the manuscript, the reader has to jump between the main body and the supporting/annex material, where hope begins to emerge that there might be something to this study beyond the vague statements typical of an opinion paper.

In conclusion, at this stage I cannot enter into specific technical recommendations because I see a profound lack of substance and consistency throughout the manuscript. I enjoyed reading it as a person, but felt utterly disappointed at the lack of substance as a scientist.

Notwithstanding my criticism, I do see potential for this paper to succeed in ESD. This is why, despite disappointed at its content after the initial excitement brought by the abstract, I do not recommend rejection at this stage. I believe in the purpose, I believe in the mission, and I hope that despite the limitations of my own review assessment, it will somehow contribute to help the authors reformulate and strengthen their message.

However, for publication to happen at ESD, it is my firm understanding that it will need to be reformulated more scientifically. And at that stage, we can discuss whether the scientific and technic details will substantiate the author's eloquent argumentation.

---

## Author Comment (AC1) · 13 Feb 2018

**R1:** The study by Baatz et al. assess the current status of the level of integration of models used by the LTER and CZO communities, and gives perspectives how the fusion of measurement and modelling communities could exploit the full strengths of observational networks to increase our understanding of Earth System Dynamics. In my opinion, this attempt is timely and deserves the presentation in a journal like Earth System Dynamics. There are many straightforward thoughts in this manuscript, and also the analysis of level of integration and the survey itself are useful instruments.

**Thank you for taking the time to review and for your positive comments on our short communication manuscript.**

**R1:** However, I see three major points that need thorough revision.

1. The utilization of terms (integration, incorporation, linkage of data model-usage, coupling) is not precise, and especially terms like model integration should be defined thoroughly and then remain reserved for the given used. One example is the abstract:
"Advancing our understanding of Earth System Dynamics (ESD) depends on the development of models and other analytical tools that integrate physical, biological and chemical data. This ambition of increased understanding and model development of ESD based on integrated site observations was at the origin of the creation of the networks of Long Term Ecological Research (LTER), Critical Zone Observatories (CZO), and others". If I understand it correctly, in the second sentence, "integrated site observations" means a set of measured variables that comprises both driving variables and target (mapped) variables that are necessary to run and validate a process based model. As the preceding sentence already uses integrated, I suggest rephrasing the next one, otherwise the readers will stumble already here. In addition, objective 3) (Line 10) suggests network integration. This needs to be specified, and distinguished from model integration. In addition, the difference between model coupling and integration is not clear, and the terms are used interchangeably (see section3.2), though they are not: Is integration of a new variable more pertinent or could these processes be used at the same time, or would model coupling be more efficient? Furthermore, it remains unclear what exactly the authors mean with data-model linkage, and how was it quantified?

**Answer: Thank you for this specific suggestion which is shared by Referee #2. We admit that some terms were used ambiguously in several places. In the revised manuscript, we define each term upon first usage and remain consistent throughout. More specifically, we use data application instead of model-data linkage and associated terms. We use steering synergies of observation networks when we refer to the 'coupling' or 'integration' of observation networks. We use more specific and exact wording for model integration and model coupling. We explain the elements at first appearance, near the conceptual diagram that present the different steps and processes for which this paper brings new perspectives.**

**R1:** 2. To this end, I suggest introducing a figure on how data assimilation, data integration and model coupling are related, where can they complement each other, and what are the differences?

**Answer: Thank you, we agree that a visualization of the concepts and processes through a flow chart would be useful. In the revised manuscript, we added the Figure below.**

[Figure]

**Figure: Flowchart of concepts, pathways and processes of applying terrestrial earth observatory network data to earth system dynamics models; identifying the three challenges of (I) data application, (II) model integration, and (III) steering synergies in observation networks.**

**R1:** 3. The structure of the paper needs to be changed. A materials and methods section is required to give the reader an overview on the questions posed in the survey, and on the evaluation methods. The current Appendix structure (Appendix figures were missing) produces duplicate information which could be avoided when materials and methods section would follow the introduction.

**Answer: Thank you for the suggestion. Your suggestion to bring the appendix into the main part of the manuscript was also made by Referee #2. In the revision of the manuscript, we follow your suggestion to bring information on Materials and Methods from the appendix into the main body of the manuscript.**

**R1:** In general, the different paragraphs even in a section or sub section need to be linked in a more straightforward way, and the writing styles need to be harmonized.

**Answer: In the revision of the manuscript, we specifically work on linking the paragraphs, one by one, in a harmonized manner to make the writing more consistent and easy to follow.**
**See for instance at page 3 L17, where we added: 'For these purposes, the '.**

**R1:** See some more detailed comments below.
Title
ok

Highlights
-
Abstract

L6: "we look for" sounds colloquial, I suggest changing to "The survey results identified gaps . . ."

**Answer: Agreed. Changed as suggested.**

L10: please specify complementarity. Unclear if you mean complementarity of networks or observations that occur at given sites, but are missing at the other.

**Answer: We use complementarity to refer to both network topology and specific elements in terrestrial observation networks. We clarify this in the revised version: 'complementarity in measurement data and spatial spread'**

L13: functional topological network: Formulation unclear. Please specify.

**Answer: Agreed. We now use a more clear formulation: '1) making the existing site-determined networks also spatially arranged networks by creating new grid-based, or random-stratified spatial network structures…'**

Introduction
L20: this is an odd sentence, I don't think it's verbatim. What exactly do you refer to with "scientific and societal imperatives" which are paramount to improve our understanding? Please rephrase the sentence, and probably split it.

**Answer: Scientific and societal imperatives refer to global change and the threats that it poses to Earth's habitability. We restructured the first three lines, and split the sentence as suggested:**

**'Complex interactions among rock, soil, water, air, and living organisms regulate the natural habitat and determine the availability of life-sustaining resources for human well-being (MEA, 2005). In the light of accelerating global change (e.g. IPCC, 2014; Camill, 2010) and safeguarding the Earth as habitable space, the scientific and societal imperatives ask to advance our understanding of Earth System Dynamics (ESD).'**

L30: exemplary is ambiguous here. I suggest to change to "are for instance networks . . . "
**Answer: Agreed, changed as suggested.**

P3L2: ESD has been introduced before.
**Answer: Yes, statement removed.**

P3L3: please rephrase sentence, it starts with ". . .range form . . .", but there is not "to".
**Answer: Right, we rephrased the sentence.**

P3L2-10: This section is fundamental in this contribution. It contains data assimilation, an attempt to define integrated/coupled models and calls for "more" integrated models.
The authors need to clearly define the borders what can be achieved with data assimilation, model integration and model coupling, work out what the differences are and to comment on if "more

integrated" models are less or more efficient than coupled models. I also suggest having a figure here on how data assimilation, model integration and model coupling is connected or distinguished.

**Answer: We think it is a great idea to add a figure to clarify these concepts. As outlined above, we expanded this paragraph, and added a figure linking concepts, connections and bottlenecks.**

L11: how exactly are more integrated and coupled models different? For more integration, capabilities of a usually couples model need to be implemented in the parent (less) integrated model. There are also several ways of coupling, which should here described in detail.

**Answer: We do prefer not to go into all details of different techniques and nuance of differentiating integration and coupling of models. We expanded this section by 1-2 sentences describing the ways of integration/coupling:**

**'An improved understanding of interactions and feedbacks among water, energy and weathering cycles with biota, ecosystem functions and services guides the development of more integrated terrestrial Earth system models (Vereecken et al., 2016b). For this reason, developing integrated models dealing with different processes, such as Land Surface Models, or coupling existing process models in suites (e.g. Peckham et al., 2013; Duffy et al., 2014) are options to expand our current modelling capability to incorporate cross-disciplinary processes for improved prediction of whole ESD system-level understanding'**

L13: At this point the authors bring in another terminology: incorporation of measurements. I suppose, this is in no way different from data integration. Please be very clear at this point. Incorporation also contradicts the coupling approach promoted in the preceding sentence.

**Answer: We now use data application throughout the manuscript. Here too, 'incorporation of measurements' was changed to 'data application'.**

L17, and following two paragraphs come out of the blue. A connecting, introducing sentence is required at this point.
**Answer: We agree and added a connecting introductory phrase: 'For these purposes, the '.**

P4L6-9: The sentence seems too long for the part of the message I understand. I am not sure what your mean with organizing science questions. If I get it right, I suggest: "At the same time, the science questions, observed variables and associated measurement methods lead to . . ."

**Answer: Thank you, we added the suggestion and shortened the sentence to focus more on the message.**

L10: I suggest deleting the "Here".
**Answer: Agreed. We used "For this study," instead.**

L12: still at this point model integration and linkage of data-model usage remain vague.
**Answer: We followed your suggestion, defined the terms 'model integration' and 'data application' earlier in the manuscript (as outlined above), and use them consistently here.**

L17-28: Introduction could stop here. I find it very odd that you dive in section 2 directly into the Challenges without describing what was asked in the survey. I strongly advocate for a materials and methods section, as the neglect of the named section leads to duplicate information in the appendix. **Answer: We agree and, integrated this section with the newly introduced Materials and Methods section.**

Materials and Methods (in appendix, unfortunately)
**As suggested, we introduced Materials and Methods now by presenting the survey.**

Results
P5L1: How is data-model linkage defined? This needs to be explained in a preceding section (e.g., Materials methods or such like) P5L3: Appendix and supplement: quite vague. All appendix, all supplements? The fundamental data should occur in the main text body.

**Answer: Thank you. We addressed the previous two comments by following your suggestion to place the Materials and Methods section between the Introduction and the Challenges in the main body of the paper.**

L15: Does "used" mean that it is an input variable or a target variable. Please consider having the categories input and target variables at this point. It may be useful information if calibration/validation data has been determined at the site directly.

**Answer: In our survey we asked about usage of the variables used as 'model input,' 'input or calibration/validation', or both. We now moved the following sentence to the more relevant position at the end of this paragraph:**
**'The average model used 14 variables of the supplied list, ~2/3 of the variables for model input and ~1/3 for calibration/validation (Appendix** Error! Reference source not found.**).'**

L20: I don't see the relation of this paragraph to the section head "data-model linkage".
Please consider having another section head, or link figure one with (for instance) scales of the input data. This would be the link to your section head. In Figure 1 I suggest leaving the 0-values white.

**Answer: We revised this text to clarify its meaning and emphasized its relevance to Figure 1 (now Figure 2). This figure delineates the character of current model usage and observations within the LTER and CZO networks based on results from the extensive survey. To make this more clear we added: 'This result stresses the relevance of both observational networks to ESD processes in terms of the spatial and temporal scales in CZO/LTER modelling activities. At the same time, Figure 2 reveals a lack of modeling activities at the larger scales (continental and global scales).'**
**From our perspective and as stated, linking CZO/LTER observation data to ESD models is critical to addressing these scaling challenges.**

**Regarding the 0-values of Figure 2, we left the 0-values light-grey, because the survey relies on participants' answers and is not exclusive. Thus, the light-grey color stresses the light/dark contrast but not the values.**

P6, section 2.2: I suggest reducing this section to one paragraph, i.e., only giving one example, but explain this properly. The corresponding figure is illegible, especially the river-like structure, the model structure? (seed rain model and some ellipses and arrows, without explanation), is it boxplots I am looking at, is the reference scenario "human impacted" only? Also after several readings it is not clear to

me what the difference between data-model linkage and integrated modelling should be, especially if the Figure caption starts with "Integrated model . . ." and section 3 is on integrated modelling. The reference to the figure is missing.

**Answer: We clarified this figure by reducing it to its key elements. We highlighted the relevance of the example to the Challenge 1 by stressing the huge number of diverse and site-specific types of observational data used in current LTER/CZO models:**

**'Next to the common, cross-site measurements, CZO and LTER datasets generally include site-specific types of observations gathered to answer site-specific scientific questions on model development, ecosystem response to global change and prediction.'**

[Figure]

**New caption Figure 3: LTER sites answer specific ecological questions, for which specific data are gathered, e.g., Black poplar population persistence under climate change. Black poplar population strength along the Loire River section (top), and model projections under current, climate change and adaptation management scenarios (Van Looy and Piffady, 2017).**

L24 "holistic integrated models". This class has not been defined before. Does this class comprise all the models used by the survey respondents?
**Answer: We agree that the term holistic does not introduce new information, and therefore removed this term.**

L25/26: unclear what "replaced" means. Would modellers prefer using other products than the directly measured ones, and what would be the reason? The following sentence claims that there is a need for on-site measurements, but would they be used?

**Answer: Agreed. The term replaced is ambiguous. We reformulated:**

**'As the survey results have shown, for some themes (e.g., habitat/vegetation/crop) remotely sensed or existing database information was preferably used (Appendix** Error! Reference source not found.**) in contrast to potential data from on-site field measurements.'**
**Furthermore, we argue that generally on-site measurements are more accurate than remotely sensed information.**

P7L9-10: All disciplines and compartments should be subject to the main text.
Subsection 3.1: I don't fully understand how the level of integration was calculated exactly, and how the variables (L12, 14) find their way into this metric. Please specify. The metric "level of integration" is also not discussed in this section, but there is only a reference to the figure.

**Answer: We agree that the level of integration needs to be clear. We moved our explanation of this metric from the appendix to the main text:**
**'The model 'level of integration' was calculated by normalizing the model-wise number of disciplines and compartments indicated by the responses to a scale of 0 to 1 (high) and equally weighing both to obtain the 'level of integration' between 0 and 1.'**

L30-31: According to the subsection head this section is about integrated modelling. In the named lines, it says "requires coupling of models. . .". This switch of model integration to coupling is really confusing, and inadequate.

**Answer: We agree that it is confusing to use coupling and integration interchangeably. We now stick to the one term integration – that includes the 'coupling' of models through model couplers - for the revision and modified the passage accordingly:**
**'Here we show the example of RT-FluxPIHM, which integrates processes of reactive transport (RT) model with a land-surface and hydrologic model (FluxPIHM) (Figure 5) (Bao et al., 2017;Li et al., 2017a).'**

P8L1: RT-FluxPIHM: what is the purpose of this model?

**Answer: Right, we clarified that this model integrates geochemistry and hydrology:**
**'…RT-FluxPIHM, which integrates processes of reactive transport (RT) model with a land-surface and hydrologic model (FluxPIHM)'**

L10: I suggest stopping here (don't give the next example) and tell the reader what the preliminary results were. The figure suggests increased groundwater flow though now trees with a greater rooting depth are present. Please discuss this briefly, and what regional implications does it have?

**Answer: We agree and omitted the second example. We clarified that the examples are presented not for their results, but to illustrate how the models are constructed. References are provided if the reader is interested in the specific results. We briefly discuss the results now in the text:**
**'The enhanced vertical macro-pore development through deeper roots of woody encroachment compared to grass led to higher groundwater flow (Figure 5b).'**

P9L18: please add the appropriate reference to your analysis.
**Answer: The reference to the Figure was added (**Error! Reference source not found.**).**

P10L18-20: Please rephrase sentence.
**Answer: The sentence was rephrased to be more clear:**
**'**

P11L18-20: True, but this paper would be a chance to describe how this could be set into action.
**Answer: Agreed, but we regret that describing frameworks is out of the scope of this paper.**
**We rephrase:**
**'Measurements of biota in the subsurface (e.g., bacteria, fungi, roots), especially at depth, is expecting strong developments in the coming decade as is the modelling (Grandy et al., 2016). Developing method for harmonization and standardization of biotic observations will better facilitate access to biotic observations related to processes and feedbacks with the hydrologic and geochemical cycles. In turn, stronger steering of synthesis between CZO and LTER would feed the LTER need for a deeper geoscience emphasis.'**

P12L29-30: Please finish this sentence.
**Answer: Right, the last word "gradient" was added in a revision.**

This section: what do the communities think of your two suggestions? Would a new network replace the old ones? Please also specify the term functional topological networks. What does it imply for the networks.

**Answer: Thank you, these are valid questions.**
**We clarified what we meant with topological networks:**
**'The integration of existing research and observation networks can be accomplished either by 1) spatially arranging the networks by creating new grid-based, or random-stratified spatial network structures (Brantley et al., 2017), or by 2) thematically (e.g., landscape processes, human impact) and geographically (e.g., climate, altitude) restructuring the available networks. '**
**Part of the answer to your question is given in the following two paragraphs starting with "As indicated above, the situation of LTER and CZO research as well as…"**
**We emphasize, that there is not one solution shared by all stakeholders:**
**'The challenges ahead for US-LTER, US CZO and NEON are as specific as those for the European efforts to establish one joint research infrastructure.'**
**The development of networks needs to be understood as a process. This is addressed and reflected in the paragraph starting with:**
**'Some existing Earth observation networks …'**
**Continued with:**
**'A notable European initiative is…'**

P13L19: here the term "more deeply-informed" comes into play as another model class. It is necessary for the authors to clarify how this translates to model coupling, integration in their opinion.
**Answer: We agree that this may cause confusion. We avoided using the term 'deeply-informed' when talking about models informed by data/data assimilation and use the terminology that has already been introduced.**

---

## Author Response (AR1)

**R1:** The study by Baatz et al. assess the current status of the level of integration of models used by the LTER and CZO communities, and gives perspectives how the fusion of measurement and modelling communities could exploit the full strengths of observational networks to increase our understanding of Earth System Dynamics. In my opinion, this attempt is timely and deserves the presentation in a journal like Earth System Dynamics. There are many straightforward thoughts in this manuscript, and also the analysis of level of integration and the survey itself are useful instruments.

**Thank you for taking the time to review and for your positive comments on our short communication manuscript.**

**R1:** However, I see three major points that need thorough revision.

1. The utilization of terms (integration, incorporation, linkage of data model-usage, coupling) is not precise, and especially terms like model integration should be defined thoroughly and then remain reserved for the given used. One example is the abstract:
"Advancing our understanding of Earth System Dynamics (ESD) depends on the development of models and other analytical tools that integrate physical, biological and chemical data. This ambition of increased understanding and model development of ESD based on integrated site observations was at the origin of the creation of the networks of Long Term Ecological Research (LTER), Critical Zone Observatories (CZO), and others". If I understand it correctly, in the second sentence, "integrated site observations" means a set of measured variables that comprises both driving variables and target (mapped) variables that are necessary to run and validate a process based model. As the preceding sentence already uses integrated, I suggest rephrasing the next one, otherwise the readers will stumble already here. In addition, objective 3) (Line 10) suggests network integration. This needs to be specified, and distinguished from model integration. In addition, the difference between model coupling and integration is not clear, and the terms are used interchangeably (see section3.2), though they are not: Is integration of a new variable more pertinent or could these processes be used at the same time, or would model coupling be more efficient? Furthermore, it remains unclear what exactly the authors mean with data-model linkage, and how was it quantified?

**Answer: Thank you for this specific suggestion which is shared by Referee #2. We admit that some terms were used ambiguously in several places. In the revised manuscript, we define each term upon first usage and remain consistent throughout. More specifically, we use data application instead of model-data linkage and associated terms. We use steering synergies of observation networks when we refer to the 'coupling' or 'integration' of observation networks. We use more specific and exact wording for model integration and model coupling. We explain the elements at first appearance, near the conceptual diagram that present the different steps and processes for which this paper brings new perspectives.**

**R1:** 2. To this end, I suggest introducing a figure on how data assimilation, data integration and model coupling are related, where can they complement each other, and what are the differences?

**Answer: Thank you, we agree that a visualization of the concepts and processes through a flow chart would be useful. In the revised manuscript, we added the Figure below.**

[Figure]

**Figure 1: Flowchart of concepts, pathways and processes of applying terrestrial observatory network data to earth system dynamics models; identifying the three challenges of (I) data application, (II) model integration, and (III) steering synergies in observation networks.**

**R1:** 3. The structure of the paper needs to be changed. A materials and methods section is required to give the reader an overview on the questions posed in the survey, and on the evaluation methods. The current Appendix structure (Appendix figures were missing) produces duplicate information which could be avoided when materials and methods section would follow the introduction.

**Answer: Thank you for the suggestion. Your suggestion to bring the appendix into the main part of the manuscript was also made by Referee #2. In the revision of the manuscript, we follow your suggestion to bring information on Materials and Methods from the appendix into the main body of the manuscript.**

**R1:** In general, the different paragraphs even in a section or sub section need to be linked in a more straightforward way, and the writing styles need to be harmonized.

**Answer: In the revision of the manuscript, we specifically work on linking the paragraphs, one by one, in a harmonized manner to make the writing more consistent and easy to follow.**

**See for instance at page 3 L15, where we added:** '**For these purposes, the** '.

**R1:** See some more detailed comments below.
Title
ok
Highlights
-
Abstract

L6: "we look for" sounds colloquial, I suggest changing to "The survey results identified gaps . . ."

**Answer: Agreed. Changed as suggested.**

L10: please specify complementarity. Unclear if you mean complementarity of networks or observations that occur at given sites, but are missing at the other.

**Answer: We use complementarity to refer to both network topology and specific elements in terrestrial observation networks. We clarify this in the revised version:**
**'identifying complementarity in measured variables and spatial extent'**

L13: functional topological network: Formulation unclear. Please specify.

**Answer: Agreed. We now use the following formulation in the revised version:**
**'including co-location of sites of the existing networks and further formalizing these recommendations among these communities.'**

Introduction
L20: this is an odd sentence, I don't think it's verbatim. What exactly do you refer to with "scientific and societal imperatives" which are paramount to improve our understanding? Please rephrase the sentence, and probably split it.

**Answer: Scientific and societal imperatives refer to global change and the threats that it poses to Earth's habitability. We restructured the first three lines, and split the sentence as suggested:**

**'Complex interactions among rock, soil, water, air, and living organisms regulate the natural habitat and determine the availability of life-sustaining resources for human well-being (MEA, 2005). In the light of accelerating global change (e.g. Camill, 2010; IPCC, 2014) and safeguarding the Earth as habitable space, the scientific and societal demands require improved understanding of Earth System Dynamics (ESD).'**

L30: exemplary is ambiguous here. I suggest to change to "are for instance networks . . . "
**Answer: Agreed, changed as suggested.**

P3L2: ESD has been introduced before.
**Answer: Yes, statement removed.**

P3L3: please rephrase sentence, it starts with ". . .range form . . .", but there is not "to".

**Answer: Right, we rephrased the sentence.**

P3L2-10: This section is fundamental in this contribution. It contains data assimilation, an attempt to define integrated/coupled models and calls for "more" integrated models.
The authors need to clearly define the borders what can be achieved with data assimilation, model integration and model coupling, work out what the differences are and to comment on if "more integrated" models are less or more efficient than coupled models. I also suggest having a figure here on how data assimilation, model integration and model coupling is connected or distinguished.

**Answer: We think it is a great idea to add a figure to clarify these concepts. As outlined above, we expanded this paragraph, and added a figure linking concepts, connections and bottlenecks.**

L11: how exactly are more integrated and coupled models different? For more integration, capabilities of a usually couples model need to be implemented in the parent (less) integrated model. There are also several ways of coupling, which should here described in detail.

**Answer: We do prefer not to go into all details of different techniques and nuance of differentiating integration and coupling of models. We expanded this section by 1-2 sentences describing the ways of integration/coupling:**

**'Integrated models include cross-scale and cross-disciplinary processes that are needed to fully predict ESD responses to perturbations from driving forces at local to global scales. No single ESD model can accomplish the full representation of driver and response functions. For this reason, developing integrated models dealing with different processes, such as Land Surface Models, or coupling existing process models in suites (e.g. Duffy et al., 2014; Peckham et al., 2013) are options to expand our current modelling capability to incorporate cross-disciplinary processes for improved prediction of whole ESD system-level understanding, as well as for policy and management decisions.'**

L13: At this point the authors bring in another terminology: incorporation of measurements. I suppose, this is in no way different from data integration. Please be very clear at this point. Incorporation also contradicts the coupling approach promoted in the preceding sentence.

**Answer: We now use data application throughout the manuscript. Here too, 'incorporation of measurements' was changed to 'data application'.**

L17, and following two paragraphs come out of the blue. A connecting, introducing sentence is required at this point.
**Answer: We agree and added a connecting introductory phrase: 'For these purposes, the '.**

P4L6-9: The sentence seems too long for the part of the message I understand. I am not sure what your mean with organizing science questions. If I get it right, I suggest: "At the same time, the science questions, observed variables and associated measurement methods lead to . . ."

**Answer: Thank you, we added the suggestion and shortened the sentence to focus more on the message.**

L10: I suggest deleting the "Here".
**Answer: Agreed. We used "For this study," instead.**

L12: still at this point model integration and linkage of data-model usage remain vague.
**Answer: We followed your suggestion, defined the terms 'model integration' early in the manuscript (as outlined above), and use the term consistently.**
**New formulation: 'Integrated models include cross-scale and cross-disciplinary processes that are needed to fully predict ESD responses to…'**
**We also use the term 'data application' to models now instead of data-model usage or linkage, or related expressions to be more consistent and clearer.**

And
L17-28: Introduction could stop here. I find it very odd that you dive in section 2 directly into the Challenges without describing what was asked in the survey. I strongly advocate for a materials and methods section, as the neglect of the named section leads to duplicate information in the appendix.
**Answer: We agree and, integrated this section with the newly introduced Materials and Methods section.**

Materials and Methods (in appendix, unfortunately)
**We introduced a Materials and Methods now by briefly presenting the survey setup and analysis of the responses. The results which were previously in the appendix, were now integrated into the 'Current status' sub sections of the three challenges.**

Results
P5L1: How is data-model linkage defined? This needs to be explained in a preceding section (e.g., Materials methods or such like) P5L3: Appendix and supplement: quite vague. All appendix, all supplements? The fundamental data should occur in the main text body.

**Answer: Thank you. We addressed the previous two comments by following your suggestion to place the Materials and Methods section between the Introduction and the Challenges in the main body of the paper.**

L15: Does "used" mean that it is an input variable or a target variable. Please consider having the categories input and target variables at this point. It may be useful information if calibration/validation data has been determined at the site directly.

**Answer: In our survey we asked about usage of the variables used as 'model input,' 'input or calibration/validation', or both. We now moved the following sentence to the more relevant position at the end of this paragraph:**
**'The average model used 14 variables of the supplied list of 52, ~2/3 of the variables for model input and ~1/3 for calibration/validation (Table A1).'**

L20: I don't see the relation of this paragraph to the section head "data-model linkage".
Please consider having another section head, or link figure one with (for instance) scales of the input data. This would be the link to your section head. In Figure 1 I suggest leaving the 0-values white.

**Answer: We revised this text to clarify its meaning and emphasized its relevance to Figure 1 (now Figure 2). This figure delineates the character of current model usage and observations within the LTER and CZO networks based on results from the extensive survey. To make this more clear we added:**

'This result stresses the relevance of both observation networks to ESD processes in terms of the spatial and temporal scales in CZO/LTER modelling activities. At the same time, Figure 2 reveals a lack of modelling activities at larger extent (continental and global) in both communities.'
From our perspective and as stated, linking CZO/LTER observation data to ESD models is critical to addressing these scaling challenges.

Regarding the 0-values of Figure 2, we left the 0-values light-grey, because the survey relies on participants' answers and is not exclusive. Thus, the light-grey color stresses the light/dark contrast but not the values.

P6, section 2.2: I suggest reducing this section to one paragraph, i.e., only giving one example, but explain this properly. The corresponding figure is illegible, especially the river-like structure, the model structure? (seed rain model and some ellipses and arrows, without explanation), is it boxplots I am looking at, is the reference scenario "human impacted" only? Also after several readings it is not clear to me what the difference between data-model linkage and integrated modelling should be, especially if the Figure caption starts with "Integrated model . . ." and section 3 is on integrated modelling. The reference to the figure is missing.
**Answer: We clarified this figure by reducing it to its key elements. We highlighted the relevance of the example to the Challenge 1 by stressing the huge number of diverse and site-specific types of observational data used in current LTER/CZO models:**
**'Next to the common, cross-site measurements, CZO and LTER datasets generally include site-specific types of observations gathered to answer site-specific scientific questions on model development, ecosystem response to global change and prediction.'**

[Figure]

**New caption Figure 3: 'LTER sites answer specific ecological questions, for which specific data are gathered, e.g., Black poplar population persistence under climate change. Black poplar population**

strength along the French Loire River section (top), and model projections under current, climate change and adaptation management scenarios (Van Looy and Piffady, 2017).'

L24 "holistic integrated models". This class has not been defined before. Does this class comprise all the models used by the survey respondents?
**Answer: We agree that the term holistic does not introduce new information, and therefore removed this term.**

L25/26: unclear what "replaced" means. Would modellers prefer using other products than the directly measured ones, and what would be the reason? The following sentence claims that there is a need for on-site measurements, but would they be used?

**Answer: Agreed. The term replaced is ambiguous. We reformulated:**
**'As the survey results demonstrated, for some themes (e.g., habitat/vegetation/crop), remotely sensed or existing database information was preferably used in contrast to potential data from on-site field measurements (Table A1).'**
**Furthermore, we argue that generally on-site measurements are more accurate than remotely sensed information.**

P7L9-10: All disciplines and compartments should be subject to the main text.
Subsection 3.1: I don't fully understand how the level of integration was calculated exactly, and how the variables (L12, 14) find their way into this metric. Please specify. The metric "level of integration" is also not discussed in this section, but there is only a reference to the figure.

**Answer: We agree that the level of integration needs to be clear. We moved our explanation of this metric from the appendix to the main text on page 5 lines 5 to 7:**
**'The model 'level of integration' (an index ranging from 0 [low] to 1 [high]) was calculated by normalizing the model-wise number of disciplines and compartments indicated by the responses to a scale of 0 (none) to 1 (many) and averaging these two indices.'**

L30-31: According to the subsection head this section is about integrated modelling. In the named lines, it says "requires coupling of models. . .". This switch of model integration to coupling is really confusing, and inadequate.

**Answer: We agree that it is confusing to use coupling and integration interchangeably. We now stick to the one term integration – that includes the 'coupling' of models through model couplers - for the revision and modified the passage accordingly:**
**'Here we show the example of RT-FluxPIHM, which integrates processes from a reactive transport (RT) with a land-surface and hydrologic model (FluxPIHM) (Figure 5) (Bao et al., 2017; Li et al., 2017).'**

P8L1: RT-FluxPIHM: what is the purpose of this model?

**Answer: Right, we clarified that this model integrates geochemistry and hydrology:**
**'…RT-FluxPIHM, which integrates processes from a reactive transport (RT) with a land-surface and hydrologic model (FluxPIHM) …'**

L10: I suggest stopping here (don't give the next example) and tell the reader what the preliminary results were. The figure suggests increased groundwater flow though now trees with a greater rooting depth are present. Please discuss this briefly, and what regional implications does it have?

**Answer: We agree and omitted the second example. We clarified that the examples are presented not for their results, but to illustrate how the models are constructed. References are provided if the reader is interested in the specific results. We briefly discuss the results now in the text:**
**'The enhanced vertical macro-pore development through deeper roots of woody encroachment compared to grass led to higher groundwater flow (Figure 5b).'**

P9L18: please add the appropriate reference to your analysis.
**Answer: The reference to the Figure was added (**Fehler! Verweisquelle konnte nicht gefunden werden.**).**

P10L18-20: Please rephrase sentence.
**Answer: The sentence was rephrased to be more clear:**
**'Merging data and modelling through data assimilation also may enable testing predictions from small-scale process understanding in larger-scale, simplified model representations (Heffernan et al., 2014; Vereecken et al., 2016a).'**

P11L18-20: True, but this paper would be a chance to describe how this could be set into action.
**Answer: Agreed, but we regret that describing frameworks is out of the scope of this paper.**
**We rephrase:**
**'New initiatives have been launched recently to integrate the biotic component in Earth system science and models (Filser et al., 2016), including, for example, modelling the roles of biota (e.g., bacteria, fungi, roots) in the subsurface (Grandy et al., 2016). At the same time, improved representation of processes such as hydrologic and geochemical cycles may improve the integration of LTER models.'**

P12L29-30: Please finish this sentence.
**Answer: Right, the ending of the sentence was reformulated to:**
**'Parsimonious models can be integrated in a larger model platform (e.g. Duffy et al., 2014; Peckham et al., 2013) to investigate feedbacks over climatic and geographic gradients, and across disciplines.'**

This section: what do the communities think of your two suggestions? Would a new network replace the old ones? Please also specify the term functional topological networks. What does it imply for the networks.

**Answer: Thank you, we now removed the term 'topological' networks to avoid being ambiguous. We also reformulated the strategies proposed towards focusing more on steering synergies in the observation networks. This is reflected in the new title of the manuscript:**
**'Steering synergies in terrestrial observation networks: opportunity for advancing Earth system dynamics modelling'**
**and detailed in section 6.3 Strategies for steering synergies in Earth observatory networks. The main strategies on steering synergies of observatory networks are to increase the interaction of observatory networks, focus on data harmonization, and co-locate sites of different networks:**

'Data harmonization between networks and co-location of sites by different networks allow for more efficient allocation of resources and increases multi-compartment datasets at co-located sites. Co-location is the joint use of individual research sites by two or more networks.'
and
'Furthermore, the interaction of observatory networks increase the spatial coverage of multi-compartment observations, allowing ESD models to address research questions and testing hypothesis over larger scales gaining full benefit of multi-compartment CZO and LTER data.'

The reasons to focus on steering synergies are diverse:
'Considerations about steering observatory network synergies need to consider differences in the organizational structure, where CZOs have been mainly based on scientific networks and projects, while LTER has established formal governance structures in regional groups and globally.'
and
'In these attempts to steer synergies, the role of discussion amongst stakeholders, decision makers, funding agencies, and the broader scientific community cannot be over-stated.'

P13L19: here the term "more deeply-informed" comes into play as another model class. It is necessary for the authors to clarify how this translates to model coupling, integration in their opinion.
**Answer: We agree that this may cause confusion. We avoided using the term 'deeply-informed' when talking about models informed by data/data assimilation and use the terminology that has already been introduced.**

**Anonymous Referee #2**

The manuscript discusses a very pertinent problem in Earth System Dynamics and for that reason merits consideration in principle. However, it does so without sufficient technical coherence and depth, transmitting the notion of a vague opinion paper than a thoroughly developed scientific study, which would be needed to provide solid grounds to the argumentation conducted in the manuscript.

The obstacles being already at a scientific language level, where there is no syntax unicity behind fundamental concepts. In practice, each and every term prone to cause confusion should be clearly defined in a rigorous manner, e.g. as the other reviewer pointed out "integration, incorporation, linkage of data model-usage, coupling". Otherwise this is not a scientific paper, but rather a soft-formulated style of manuscript tailored for a non-scientific audience (which has its merits but is outside the scope of ESD).

While articulate divagations with catchy phrases and buzzwords may be popular in some soft venues in environmental science (e.g. Water Resources Research, WRR) and outreach communication (EOS or general media), this sort of approach feels out of place at ESD. In fact, here a rigorous "hard-science" treatment and discussion of problems should rather be the norm. For every bold statement of opinion, a rigorous scientific argumentation is a fundamental requirement, and in many cases mathematics are actually extremely helpful for that purpose.

My recommendation is thus to restructure and strengthen the paper so that its true scientific DNA comes out and the formulation, methodologies, definitions and overall concepts are all clearly stated. To drop the fashionable WRR-style beauty without substance, and rather take a more sober, theoretical ecology or ESD style scientific construct. The ESD readers will highly appreciate your insightful rigour.

Moreover, a clearer distinction should be made between the known facts and insights, and the novel contributions brought on by this study. The strategies followed in the study should also be more explicitly and rigorous formulated and justified, alongside with their merits and caveats.
At some stages, the reader is left to wonder what has actually been done in practice, and what are essentially suggestions and statements of purpose. The formal procedures belong to the main body of the manuscript, as they are fundamental to bring more substance to the eloquent argumentation.

In order to actually grasp some basic sense of the scientific work beneath the manuscript, the reader has to jump between the main body and the supporting/annex material, where hope begins to emerge that there might be something to this study beyond the vague statements typical of an opinion paper.

In conclusion, at this stage I cannot enter into specific technical recommendations because I see a profound lack of substance and consistency throughout the manuscript. I enjoyed reading it as a person, but felt utterly disappointed at the lack of substance as a scientist.

Notwithstanding my criticism, I do see potential for this paper to succeed in ESD. This is why, despite disappointed at its content after the initial excitement brought by the abstract, I do not recommend rejection at this stage. I believe in the purpose, I believe in the mission, and I hope that despite the limitations of my own review assessment, it will somehow contribute to help the authors reformulate and strengthen their message. However, for publication to happen at ESD, it is my firm understanding that it will need to be reformulated more scientifically. And at that stage, we can discuss whether the scientific and technic details will substantiate the author's eloquent argumentation.

We thank Referee #2 for his/her thoughtful and constructive review of our manuscript. The reviewer appears to have unmet expectations of reading a Research article, which was not the purpose of this manuscript. Rather, our manuscript is a Short Communication, describing and discussing results of a community survey to raise questions for future discussion and analysis. Bringing a high level of detail while at the same time a high level of abstraction, we thoroughly analyzed the survey's results, discussing the merits of the survey and the limitations of the analysis. The survey's outcome is described, synthesized and discussed in terms of information on synergies, challenges and perspectives to ESD modeling as seen by scientists of the CZO, LTER, NEON and ISMC networks. We do not understand the reviewers concerns about Water Resources Research, which is a fine journal publishing highly detailed and rigorous results from observational, theoretical, experimental and modeling studies. More fundamentally, we argue that our Short Communication falls squarely within the domain of ESD which is "dedicated to the publication and public discussion of studies that take an interdisciplinary perspective of the functioning of the whole Earth system and global change" and "accepts research articles, review articles, short communications, and commentaries."

Many of the Referee #2's comments were consistent with those of Referee #1. We have made our use of terms more consistent and clear and have moved the materials and methods from the appendix to the main text. We point out the difference between integrating models and coupling models, e.g. here:
'For this reason, developing integrated models dealing with different processes, such as Land Surface Models, or coupling existing process models in suites (e.g. Duffy et al., 2014; Peckham et al., 2013) are options to expand our current modelling capability to incorporate cross-disciplinary processes for improved prediction of whole ESD system-level understanding, as well as for policy and management decisions.'

Where appropriate, we reformulated and strengthened statements and argumentation, adding e.g.:
'This result stresses the relevance of both observation networks to ESD processes in terms of the spatial and temporal scales in CZO/LTER modelling activities.'
or:
'LTER sites answer specific ecological questions, for which specific data are gathered, e.g., Black poplar population persistence under climate change.'

With a conceptual graphical presentation of the flow between observation network data and model development, indicating the different aspects of data application, model integration, and steering synergies of observational networks, we believe the paper will grow to the strength and scientific DNA the reviewer refers to. We inserted:

[revised manuscript text omitted]